# Non-Anthocyanin Compounds in Minority Red Grapevine Varieties Traditionally Cultivated in Galicia (Northwest Iberian Peninsula), Analysis of Flavanols, Flavonols, and Phenolic Acids

**DOI:** 10.3390/plants12010004

**Published:** 2022-12-20

**Authors:** Ángela Díaz-Fernández, Emilia Díaz-Losada, Anxo Vázquez-Arias, Anna Puig Pujol, Daniel Moreno Cardona, María Esperanza Valdés-Sánchez

**Affiliations:** 1Research Station of Viticulture and Enology of Galicia (EVEGA), 32419 Ourense, Spain; 2Catalan Institute of Vine and Wine—Institute of Agrifood Research and Technology (INCAVI-IRTA), 08720 Vilafranca del Penedès, Spain; 3Center for Scientific and Technological Research of Extremadura—Food and Agriculture Technology Institute of Extremadura (CICYTEX-INTAEX), Avenue Adolfo Suárez s/n, 06007 Badajoz, Spain

**Keywords:** *Vitis vinifera* L., non-anthocyanin compounds, phenolic fingerprint, flavanols, flavonols, hydroxycinnamic acids, hydroxybenzoic acids, varietal differentiation

## Abstract

Non-anthocyanin compounds (NAN) such as flavonol, flavanol, and phenolic acids should be considered in the characterization of minority red grapevine varieties because these compounds are involved in copigmentation reactions and are potent antioxidants. Sixteen NAN were extracted, identified, and quantified by High Performance Liquid Chromatography (HPLC) from grapes of 28 red genotypes of *Vitis vinifera* L. grown in Galicia (Northwest of Spain) in 2018 and 2019 vintages. The percentage of total NAN with respect to the total polyphenol content (TPC) values was calculated for each sample and established into three categories: high percentage NAN varieties (NANV), those varieties showing low percentages of NAN (ANV), and finally those varieties showing medium percentages of NAN (NANAV). ‘Xafardán’ and ‘Zamarrica’, classified as NANAV, had high values of TPC and showed good percentages of flavonol and flavanol compounds. Principal component analyses (PCA) were performed with flavonol, flavanol, and phenolic acid profiles. The flavonol and flavanol profiles allowed a good discrimination of samples by variety and year, respectively. The flavonol profile should therefore be considered as a potential varietal marker. The results could help in the selection of varieties to be disseminated and in the identification of the most appropriate agronomic and oenological techniques that should be performed on them.

## 1. Introduction

Grapes are an important source of polyphenolic compounds, secondary metabolites which perform different functions. Some compounds also have a protective function against climate change effects and could even mediate induction responses to counteract pathogens [1]. 

Polyphenolic compounds contribute to sensory characteristics such as color, flavor, astringency, and hardness of wine directly or by interaction with proteins, polysaccharides, or other phenolic compounds [2,3]. The phenolic compounds of grapes can differ in flavonoid compounds such as anthocyanins (AN), flavonols (FLAVO), fla-van-3-ols (FLAVA), and non-flavonoids compounds such as hydroxybenzoic (HBA) and hydroxycinnamic (HCA) acids and their esters, stilbenes, and volatile phenols [4].

Color, an important factor for evaluating the quality of red wine, is linked to the ac-cumulation of anthocyanins (AN) in the grape berry skin. However, it is not only the anthocyanin concentration and profile that is responsible for wine color; copigmentation phenomena can account from 30% to 50% of color in young wines [5]. Copigmentation in wine results from molecular interactions between anthocyanins and other organic molecules, called cofactors, that form molecular associations or complexes. The most common cofactors include phenolic acids, flavonoids, and particularly derivatives of flavonol and flavone subgroups [6]. Previous research have focused on the most abundant copigments occurring in red wines, i.e., the monomeric (catechin and epicatechin), oligomeric (dimeric and trimeric B type proanthocyanins), and polymeric (tannins) and evaluated their effects on copigmentation and the formation of new anthocyanin-derived red wine pigments [7,8,9]. Darías-Martín et al. [8] concluded that red wine color is generally limited by the levels of cofactors available for copigmentation rather than by the level of anthocyanins alone. This demonstrates the importance of non-pigmented composition in establishing red wine color and the influence of the initial must composition; for this reason, phenolic compounds are often classified into anthocyanins and non-anthocyanins compounds. 

On the other hand, the antioxidant activity of plant polyphenols is well known. It is a well-established fact and a widely accepted concept that grapes constitute one of the most important sources of dietary polyphenolic antioxidants and the relationship of total phenolic content and different phenolic groups according to their antioxidant activity has been investigated by different research groups [10,11].

It is known that the total phenolic content and its distribution in grapes largely de-pends on varieties [9,10,12,13], terroir [14], interaction-variety-terroir [15], environmental factors [16], agronomic techniques [17,18,19,20], and their interactions [21,22].

According to Menna and Walsh [23], mature wine market countries such as Spain show a low growth in wine demand. In this context, the increasing demand for different type of wines by consumers, as well as climate change, make it necessary to recover varietal diversity. The varietal diversification could be a kind of innovation and quality sophistication of the wine product required by the sector. Minority varieties, which are those autochthonous or scarcely spread varieties that are generally linked and unique to a certain territory, and as a result are well locally adapted, should be useful tools in promoting the diversification of wine products [24]. In this regard, the assessment of the polyphenolic potential of each variety and their distribution within grapes would provide factors to evaluate the oenological potential of each variety [25]; as a result, it could optimize the oenological process and better define the type of wine that could be elaborated with each one [26]. It will be possible to plan the winemaking practices and to achieve products that differ from the rest and, therefore, to achieve greater versatility and greater interest in their potential recovery. 

On the other hand, the winemaking industry has become very important in recent times, and a careful analysis should be carried out in terms of controlling the authenticity and traceability of wines to avoid forgeries [27]. These controls are also of utmost importance in terms of guaranteeing the quality and safeness of the product [28]. They may also allow to set up new specific biomarkers for distinct grapevine varieties [29]. Among different techniques used to determine the geographic origin of wines, it could be found the one that uses phenol fingerprint [22,30,31,32,33].

Recently, we outlined the anthocyanin profile of 28 grapevine varieties, most of them minority varieties, from the northwest of the Iberian Peninsula, established in the germplasm bank of ‘Estación de Viticultura y Enología de Galicia’ (EVEGA) [12]. In that research, we established that the % AN and the % FLAVA contributed to explaining varietal differentiation, in terms of phenolic profile, for minority varieties grown in Galicia. Therefore, given the importance of the non-anthocyanin substances on the stability and intensity of the color of red wines through copigmentation phenomena and their antioxidant properties, the study of these compounds is also a major objective in the production high-quality wines, especially in terms of their color. This study is focused on (i) determining the FLAVA, FLAVO, HCA, and HCB acids profiles of 28 grapevine varieties; (ii) evaluating the usefulness of theses polyphenolic families as varietal markers; and (iii) studying the possible relationship between the varietal classification based on SSR markers and that based on their non-anthocyanin phenolic compounds characterization; that is to say, their potentiality as chemotaxonomic markers.

## 2. Results and Discussion

### 2.1. Climatic Conditions

HI was 2580.72 and 2388.88 for 2018 and 2019 vintages, respectively. This implies that the HI in the study area corresponded with warm climate (HI + 2; 2400 < HI ≤ 3000) in 2018 and warm temperate climate (HI + 1; 2100 < HI ≤ 2400) in 2019, as described in the GCCCM System [34]. CI for this location reached values from 12.93 °C (2018 vintage) to 11.80 °C (2019 vintage), characterizing the study area with cold nights (CI + 1; 12 < CI ≤ 14) to very cold nights (CI + 2; HI ≤ 12) in 2019. 

As Figure 1 reflects, the meteorological conditions did not vary strongly between 2018 and 2019. In general, 2018 was a slightly warmer and rainier season than 2019 with 14.15 °C as mean annual temperature and 1206.6 mm as mean annual accumulated rainfall, and 13.72 °C as mean annual temperature and 1064.8 mm as mean annual accumulated rainfall for both years, respectively. Thus, there were 26 days with temperatures over 35 °C in 2018 and 14 in 2019.

### 2.2. Total Polyphenols, Anthocyanin and Non-Anthocyanin Compounds

Table 1 shows the berry weight, Probable alcohol degree (PAD) and Technological maturity, as ratio between TSS (°Brix) and TA (g tartaric acid·L^−1^) of varieties investigated. The aim was to harvest the grapes at 22 °Brix (usual criteria in the region) and according to their health status. This was possible for most of the samples, with certain exceptions shown in Table 1.

Table 2 displays non-anthocyanin values (NAN), as ∑ (FLAVA, FLAVO, HCA, HBA), anthocyanin values (AN) and the total polyphenolic content (TPC) as ∑ (AN, NAN) of each variety investigated. It also shows the non-anthocyanin (% NAN) and anthocyanin percentage (% AN) with respect to the TPC. The ANOVA indicates that the effect ‘variety’ explained 82.64% of variations of NAN compounds values. With respect to the ‘year’ effect, as a general trend in most of the varieties, NAN values were higher in 2019 than in 2018 (*p* < 0.05 in 16 of them). As was reported by Díaz-Fernández et al. [12], meteorological conditions occurring in 2019 may have contributed to a better biosynthesis and accumulation of phenolic compounds in the grapes from the varieties in this area. The annual rainfalls in 2018 and 2019 were 1206.6 and 1064.8 mm, respectively, and the number of days with mean T > 35 °C were 26 in 2018 and 14 in 2019. Additionally, the grapes were smaller in 2019 than in 2018 except for ‘Brancellao’ and ‘Evega 4’ (Table 1). The relation between grape weight and phenolic substance accumulation has been investigated by several authors, and it is considered that smaller berries accumulate more phenolic compounds [35,36,37].

The highest NAN values, higher than 1000 mg·kg^−1^ FW in both years, were reached in ‘Evega 3’. In the opposite side, ‘Espadeiro’, ‘Garnacha’, ‘Gran Negro’, and ‘Mandón’ did not reach 300 mg·kg^−1^ FW any year. Values higher than 1000 mg·Kg^−1^ FW of AN in both years were registered in ‘Caiño Bravo’, ‘Castañal’, ‘Espadeiro, ‘Ferrón’, and ‘Sousón’, while ‘Evega 3’, ‘Garnacha’, ’Híbrido’, and ‘Picapoll Negro’ were varieties with low AN values, less than 200 mg Kg^−1^ FW in both years. Thus, in both years, the highest values of TPC were registered in ‘Caiño Bravo’, ‘Castañal’, ‘Corbillón’, ‘Espadeiro’, ‘Ferrón’, and ‘Sousón’, while ‘Garnacha’, ‘Híbrido’, and ‘Picapoll Negro’ were listed as the lowest phenolic varieties.

In terms of decreasing year variability, and to characterize varieties, the percentage of NAN and AN with respect to the TPC were calculated [12,38]. % NAN and %AN are shown in Table 2. As a general behavior, for a given variety, the % AN and % NAN values of 2018 were similar to those of 2019. These results allowed us to classify the investigated varieties into three categories: NANV, ANV, and NANAV as high, low, and medium percentage of NAN. Thus, in both years, ‘Corbillón’, ‘Evega 3’, ‘Evega 6’, ‘Garnacha’, ‘Híbrido’, and ‘Picapoll Negro’, varieties with values higher than 60% of NAN substances in their polyphenolic profile, should be considered as NANV. On other hand, ‘Caiño Tinto’, ‘Evega 4’, ‘Merenzao’, ‘Moscatel de Hamburgo’, ‘Pedral’, ‘Xafardán’, and ‘Zamarrica’ displayed similar percentages of NANV and ANV in both years. Finally, the remaining studied varieties could be classified as ANV varieties. The biannual means of % NAN for the ‘Mencía’ variety, widespread in Galicia, and for the ‘Tempranillo’, the most widely grown red variety in Spain, considered as a reference variety in this work, were 26.6 and 22.1%, respectively. Thus, their phenolic composition may be scarce in copigments. In this sense, ‘Caiño Tinto’, ‘Pedral’, ‘Xafardán’, and ‘Zamarrica’ had similar values of TPC to these last ones, and similar values of % NAN and AN. Thus, they could be of interest for red wine production. 

### 2.3. Flavonol Family Compounds Characterization

FLAVO are secondary metabolites that act as photo-protectors; they are largely located in the grape skins as flavonol glycosides of quercetin, kaempherol, myricetin, and isorhamnetin. They also have antioxidant properties that are released from the grape to the wine during the maceration process [39]. 

Table 3 shows FLAVO substances analyzed in grapes from the studied varieties. FLAVO were detected as 3-glucosides (Glc) of Myricetin (My), Quercetin (Quer), and Kaempferol (Kaemp), 3-rutinoside (3-Rut) of isorhamnetin (Iso), and 3-glucuronide (GlcU) of Quer. 

In 2018, My-Glc values (mg kg^−1^ FW) ranged from 0.76 in ‘Garnacha’ to 41.07 in ‘Tempranillo’, the Iso-Rut compounds varied from 0.89 in ‘Sousón’ to 25.01 in ‘Híbrido’, and the Kaemp-Glc ones from 1.32 in ‘Mouratón’ to 10.77 in ‘Merenzao’. Finally, values from 14.86 (‘Sousón’) to 87.22 (‘Zamarrica’) were recorded when the sum of quercetin compounds (∑ Quer) was evaluated. The following year, the minimum and maximum values were 3.76 and 73.37, registered in ‘Garnacha’ and ‘Sousón’, respectively, for My substances, 0.32 and 19.47 were found in ‘Zamarrica’ and ‘Tempranillo’, respectively, for the Iso ones, Kaemp compounds ranged from 0.88 in ‘Evega 3’ to 16.58 in ‘Sousón’, and values from 34.36 in ‘Moscatel de Hamburgo’ to 132.59 in ‘Xafardán’ were found for the ∑ Quer substances. Therefore, according to previous works, ‘variety’ and ‘year’, and more specifically ‘variety’, were considered as variability factors in the values registered for these compounds [40,41]. 

In this work, the factor ‘variety’ was significative and explains the highest percentage of total variation for every individual FLAVO compound (from 49.67% in Iso-Rut to 79.33% in My-Glc). On the other hand, as a general trend, every FLAVO compound except for Kaemp-Glc showed higher contents in the 2019 vintage. This effect had a significative effect in My-Glc, Quer-Glc, and Quer-Rut. The highest interannual variations were registered in My-Glc (64.70%) and Quer-Glc (61.92%). Moreover, results from Table 3 reflect that, when FLAVO compounds were analyzed on a year-by-year basis, the number of significant in-terannual differences depended on the variety considered; ‘Albarín Tinto’, ‘Caiño Bravo’, ‘Caiño Longo 2’, ‘Castañal’, ‘Corbillón’, ‘Espadeiro’, ‘Ferrón’, ‘Garnacha’, ‘Gran Negro’, ‘Mandón’, ‘Mencía’, ‘Merenzao’, ‘Moscatel de Hamburgo’, ‘Mouratón’, ‘Pan y Carne’, ‘Pedral’, ‘Picapoll Negro’, ‘Sousón’, ‘Tempranillo’, and ‘Zamarrica’ were those varieties that showed interannual significant differences for a lower number of compounds, looking to be the more stable varieties in terms of FLAVO compounds. Finally, significant effects of ‘variety x year’ were found for every flavonol compound. This effect explained a low percentage of total variation except for Iso-Rut (39.65%).

Looking at the FLAVO pattern, the main compounds in most of the analyzed samples were My and Quer compounds, and more specifically Quer-Glc, a compound that is generally higher in red-fleshed cultivars [13]. According to the results reported by Figueiredo-González et al. [42] in samples from vines also grown in Galicia, higher values of ∑ Quer than My compounds were found in ‘Mouratón’, ‘Gran Negro’, and ‘Brancellao’. Previous works have reported that these compounds were the most abundant FLAVO in the profile of red varieties [13,22,30,43].

Flavonols play critical roles in colour stabilization, as cofactors with free anthocyanins in copigmentation of young red wines, and in the evolution of complex pigments during wine aging [44,45,46] The most stable copigments associations occur between Quer compounds and the main anthocyanin, as malvidin-3-O-glucoside, in red wines [47,48]. According to Rustioni et al. [49], Quer-Glc was found to correlate with the strength of copigmentation; thus, since Quer compounds play an important role in wine copigmentation together with anthocyanins. In addition to being grape/wine bioactive compounds of possible importance for human health and nutrition, the growth of varieties with high amounts of these compounds could be considered. In this sense, ∑ Quer values of ‘Tempranillo’ were 46.28 in 2018 and 80.28 in 2019. Higher amounts than in the reference variety were registered in 2018 for ‘Mouratón’ (52.38), ‘Pedral’ (52.38), and ‘Picapoll Negro’ (49.52), in 2019 for ‘Evega 4’ (87.86), and in both years for ‘Brancellao’ (78.46 and 100.89), ‘Híbrido’ (62.53 and 106.61), ‘Xafardán’ (47.04 and 132.59), and ‘Zamarrica’ (87.22 and 99.88).

On the other hand, and considering the importance of FLAVO compounds, it has been considered of interest to examine the FLAVO profile of the samples. For this reason, and with the main objective of reducing the ‘year’ variability and thus obtaining a better characterization of the varieties, My, Iso, Kaemp, and Quer percentages with respect to their TPC were calculated for each sample and year. As Figure 2 reflects, for a given variety, similar percentages were registered in 2018 and 2019 years. Thus, ‘Mouratón’ registered values higher than 3% of My in both years. On the other hand, ‘Brancellao’, ‘Garnacha’, ‘Híbrido’, and ‘Picapoll Negro’, in both years, reached values higher than 8% of ∑ Quer. We note the high percentage of Iso (12%) found in ‘Híbrido’ in 2018. ‘Evega 4’, ‘Gran Negro’ and ‘Mandón’ had the highest values of % Kaemp (higher than 1% in all cases) in both years. 

Finally, Figure 2 highlights that ‘Zamarrica’, a variety considered of interest because of its high anthocyanin content, showed high values of % Quer (7.55 and 5.18) in 2018 and 2019, respectively.

### 2.4. Flavanol Family Compounds Characterization

In wine as well as in grapes, FLAVA are found in monomeric, dimeric, oligomeric (3 to 10 units of flavan-3-ols), and polymeric forms (more than 10 units of flavan-3-ols). FLAVA such as monomers (catechin, epicatechin and gallocatechin derivatives) and flavan-3,4-diol dimers such as procyanidins B1, B2, and B3 are present in the skin and mainly in grapes seeds. The type of flavan-3-ols in grape berries is variable between species, developmental stage, and tissue types. It is very well known that skins contain both catechins and gallocatechins and their correspondingly derived proanthocyanidins (i.e., procyanidins and prodelphinidins), whereas seeds present only catechins and procyanidins; they are secondary metabolites that trigger bitterness and astringency [50].

FLAVA are quite high reactive substances and, together with anthocyanins, would trigger the formation of different pigments and tannins in the wine-making process that modify the wine color and taste [51]. During aging, there are three main reactions in which anthocyanins participate, namely direct polymerization between anthocyanins and flavanols that produces anthocyanin-flavanol (A-T) and/or flavanol- anthocyanin (T-A) oligomers, indirect polymerization between anthocyanins and flavanols via acetaldehyde which produces purple flavanol-ethyl-anthocyanin adducts, and the formation of pyrano-anthocyanins [52]. Thus, many researchers have focused their attention on most of the flavanols found in red wines, i.e., the monomeric (catechin and epicatechin), oligomeric (dimeric and trimeric B type procyanidins), and polymeric (tannins), and they evaluated their effects on copigmentation and the formation of new anthocyanin-derived red wine pigments. 

Table 4 presents the values in mg·kg^−1^ FW of catechin (Cat), epicatechin (Epi), gallocatechin (Galo), epicatechin-gallate (GaloEpi), and the dimers PB1 and PB2 identified and quantified in grapes of different varieties from the galician EVEGA germplasm bank in 2018 and 2019 vintages. Cat in 2018 and Galo in 2019 were the major FLAVA compounds in almost every variety. The exceptions to this general behavior were ‘Ferrón’, ‘Gran Negro’, and ‘Sousón’, with PB1 as the major FLAVA in both vintages and ‘Espadeiro’ in 2018 with PB2. In general, the minority FLAVA compounds were GaloEpi in 2018, and this last one and Epi in 2019. 

Previous works reported Cat as the most important individual FLAVA in both skins and seeds [53,54] and PB1 has been reported to be the main oligomer in skins [53,55,56]. On the other hand, considering that the two-years mean ratio catechin/-epicatechin was 2.15, it would be possible to suggest that, in these varieties, the leucoanthocyanidin reductase enzyme is more active than the anthocyanidin reductase, as shown in Fanzone et al. [57] in Malbec grapes.

Since a wide range of concentrations can be observed in every compound, the ‘variety’ factor was significant and explained a big percentage of variation (>50%) in all of them except for GaloEpi. When the sum of monomers Σ (Cat, Epi, Galo, and GaloEpi), is considered in 2018, ‘Mouratón’, considered as an ANV, registered the lowest value (59.89) while ‘Corbillón’, classified as an NANV, registered the highest one (701.56 mg·kg^−1^ FW). In 2019, the values ranged from 88.08 in ‘Gran Negro’ up to 1194 in ‘Corbillón’. The minimum procyanidin total values, as Σ (PB1, PB2), corresponded to ‘Híbrido’ (30.19), and the maximum to ‘Evega 3’ (330.67), while in 2019 they corresponded to ‘Gran Negro’ (42.05) and ‘Corbillón’ (195.31), respectively. Liang et al. [43] found mean values of 31, 10, and 4 mg·kg^−1^ FW for Cat, Epi, and GaloEpi, and 9.4 and 2 for PB1 and PB2 when polyphenolic profiles in the berry samples of 344 European grape (*Vitis vinifera* L.) cultivars were evaluated after removing all seeds for two consecutive years. The grapes of cv ‘Tempranillo’, the reference variety in this work, registered values of 69.13 and 129.41 for the sum of monomers and 42.83 and 51.12 as dimers in 2018 and 2019, respectively. Thus, ‘Corbillón’ and ‘Evega 3’ were characterized by the highest FLAVA values of monomers and dimers. These facts must be considered in the winemaking process of these varieties.

On other hand, Álvarez et al. [58] reported that the prefermentative addition of cofactors as catechins and flavanols, extracted from white grape skin or seeds, increased anthocyanin copigmentation reactions and produced wines with more intense color, higher anthocyanin concentration, superior contribution of anthocyanins to the color of the wine, and less astringency. In this sense, ‘Evega 3’, a variety with low amounts of anthocyanins (<100 mg kg^−1^ FW) and high of FLAVO could be considered for this purpose. Finally, it should be emphasized that Pérez-Álvarez et al. [59] found a positive and significant correlation between antioxidant activity and total amount of FLAVA compounds.

As can be seen, except for PB2, every FLAVA compound showed higher values in 2019. ANOVA found a significant effect of the ‘year’ in Galo, GaloEpi, and sum of catechins (Σ Cat). The ‘year’ effect explained a low percentage of the variance (around 25–30%, in both compounds). On the other hand, it should be noted that different varieties were affected by the ‘year’ effect in a different way; Thus, ‘Albarín Tinto’, ‘Brancellao’, ‘Caiño Tinto’, ‘Evega 4’, ‘Evega 6’, ‘Gran Negro’, ‘Mandón’, ‘Merenzao’, ‘Pan y Carne’, ‘Pedral’, ‘Sousón’, and ‘Tempranillo’ were those varieties that showed interannual significant differences among years for a lower number of FLAVA compounds, which could initially be interpreted as more regular varieties among years. A significant effect for the interaction ‘variety x year’ was also found for the values of every compound studied, accounting for more than 34% of variation in case of GaloEpi. 

Thus, according to previous works, the amount and distribution of FLAVA are determined by genetic factors and are affected by meteorological factors [20,45,60,61,62,63]. To determine the FLAVA profiles, percentages of Cat, Epi, Galo, GaloEpi, PB1, and PB2 with respect to their TPC in 2018 and 2019 were calculated. As reflected in Figure 3, in both years, ‘Corbillón’, ‘Evega 3’, ‘Evega 6’, and ‘Xafardán’ reached values higher than 15% of Cat. Moreover, ‘Corbillón’ reached Epi values of over 10% in both years. The highest values of % Galo (> 5%) were registered in 2019 in ‘Corbillón’, ‘Evega 3’, ‘Evega 6’, ‘Garnacha’, ‘Híbrido’, ‘Xafardán’, and ‘Zamarrica’. ‘Híbrido’ and ‘Garnacha’, in 2019, were the only ones with values higher than 5% of GaloEpi (9.72% and 8.04%, respectively). It should be noted that, among monomers, Epi is considered a better copigment than Cat [7,64,65] and that galloylation at C3 of the Cat units improves the ability of FLAVA to act as copigments [65,66]. With respect to dimers, % of PB1 and PB2, ‘Evega 3’, ‘Evega 4’, and ‘Evega 6’ stood out by their high values in both years. Thus, all the varieties here mentioned were grouped as varieties rich in FLAVA compounds. Among them, we could highlight ‘Corbillón’.

### 2.5. Phenolic Acids Family Compounds Characterization

Phenolic acids exist predominantly as hydroxybenzoic (HBA) and hydroxycinnamic acids (HCA) that occur in free or conjugated form.

Table 5 shows gallic (GA) as the HBA, and caffeic (CF), ferulic (FR), and coutaric acid (COU) as HCA identified and quantified in the grape samples under study. To the best of our knowledge, this is the first time that the phenolic acid profiles of most of these varieties have been studied. Among the acids analyzed, the hydroxybenzoic acid gallic (GA) was the major compound in all varieties in both years except for ‘Albarín Tinto’ and ‘Merenzao’, where coutaric acid (COU) reached the highest values. In 2018 and 2019, the highest values of GA were registered in ‘Corbillón’ (5.88 and 6.15 mg·kg^−1^ FW), ‘Evega3’ (10.98 and 3.65), and ‘Evega 6’ (6.44 and 3.24). According to Garrido and Borges [29], GA is usually the most abundant substance of this group, and it is described as the most important phenolic compound since it is the precursor of all hydrolyzable tannins and is encompassed in condensed tannins. After it, COU was the most abundant substance belonging to phenolic acids identified and quantified, with a minimum in ‘Zamarrica’ (0.85 mg·kg^−1^ FW) and ‘Tempranillo’ (0.88) in 2018, and in ‘Brancellao’ (0.0) and ‘Zamarrica’ (0.24) in 2019. In front, ‘Evega 6’ (2.27) and ‘Pedral’ (1.66) registered the maximum values in 2018 and 2019. Finally, with respect to CF, it is noticed that there were generally higher values in 2019, ‘Caiño Longo 2’ and ‘Espadeiro’ being those varieties with maximum values. With respect to FR, it was the acid that showed the smallest quantities, with ‘Sousón’ and ‘Espadeiro’ recording the highest values in 2019. Thus, regarding to the global hydroxycinnamic compounds, the richest varieties were ‘Corbillón’ in 2018 (4.41) and ‘Sousón’ in 2019 (4.78).

As in flavonoid compounds, the effect ‘variety’ was significative for almost every compound. Several studies have shown that the phenolic content in grapes may vary according to varieties, environmental factors, and agronomic techniques [9,14,17]. According to Somers et al. [67], in the case of the genus *Vitis*, it is possible to distinguish the different varieties through the amount of p-coumaroyl and caffeoyl tartaric esters present on them, which is why they are mentioned in the literature as taxonomic markers of both grapes and wine [67]. In this work, the highest % of variation due to ‘variety’ effect was 64.53% in GA. This percentage is far below of the 93.67% found in Cat, and quite different from what happened with anthocyanin compounds in the previous study [12], in which the ‘variety’ explained as much as 93.26% of the total anthocyanin variance. With respect to the ‘year’ effect, GA and COU were generally higher in 2018 than in 2019, while the opposite happened in CF and FR. Furthermore, ‘Albarín Tinto’, ‘Brancellao’, ‘Caiño Longo 1’, ‘Castañal’, ‘Espadeiro’, ‘Ferrón’, ‘Garnacha’, ‘Híbrido’, ‘Mandón’, ‘Mencía’, ‘Merenzao’, ‘Mouratón’, ‘Pan y Carne’, ‘Picapoll Negro’, ‘Sousón’, ‘Xafardán’, and ‘Zamarrica’ were those varieties that showed interannual significant differences for a fewer number of acid compounds, which a priori could be associated with more regularity among years. There were also found significant differences for the interaction ‘variety x year’ on the values of every compound.

The short shelf life of wines of certain varieties requires careful control of their chromatic characteristics, with the copigmentation effect generally being an important tool for improving their chromatic characteristics. As mentioned above, several works have revealed that the best copigments found in red wines are FLAVO and HCA. In addition, HCA participates in the formation of new anthocyanin-derived pigments in wine, known as hydroxyphenyl-pyranoanthocyanins. Moreover, GA seems to be stable along the wine aging process [68], and the prefermentative addition of GA at appropriate levels might be a promising enological technology to obtain wines with high color quality and aging potential [68,69]. In this sense, it is important to highlight that varieties such as ‘Evega 3’, ‘Evega 6’, and ‘Corbillón’ showed higher quantities in both years than the mean content found in the ‘Tempranillo’ reference variety, grown in the same edaphoclimatic conditions. In addition, relevant research has indicated that covinification could make the color of red wines more stable, since the copigmentation and anthocyanin self-association processes occur more favorably in covinification wines than in their mono-varietal counterparts [70], so the growth and use of some of these varieties could be considered. However, special attention must be paid during the winemaking and storage processes because wines with high concentrations of hydroxycinnamic acids (especially COU and FR) can develop strong phenolic flavors in the presence of *Brettanomyces* yeasts through the decarboxylation of hydroxystyrenes, which are subsequently reduced to ethyl phenols [71].

The percentages of each phenolic acid with respect to the TPC were calculated and Figure 4 reflects the phenolic acids profiles of varieties analyzed. ‘Corbillón’, ‘Evega 3’, ‘Merenzao’, ‘Moscatel de Hamburgo’, and ‘Picapoll Negro’ varieties showed the highest % of GA, all of them in 2018. ‘Garnacha’, ‘Híbrido’, and ‘Merenzao’ displayed those highest of COU, also in 2018.

### 2.6. Discrimination of Varieties Classification

To better understand the potential of these families of phenolic compounds as a tool for varietal classification, as well as to classify the varieties under study, the results obtained were subjected to PCAs analysis

To determine their potential as chemotaxonomic tools of FLAVO, FLAVA, and AC (HCA + HBA) families, a first PCA was performed with their respective global percentage values with respect to TPC.

Figure 5 shows that F1 and F2 accounted 83.72% of the total variance (55.87% and 27.85%, respectively). In general, samples from the same variety and different years were found in the same group. F1 separated two groups and a subgroup of varieties according to their % FLAVO, % HBA and HCA, and % FLAVA values. In accordance with the PCA results, contributions of each family were: AC in a 42%, followed by FLAVA with 36% and FLAVO with 22% to F1, while FLAVO contributes 72% to F2, followed by FLAVA with 26% and AC with 2%. For this reason, FLAVO seems to be the most influential family in the varieties’ differentiation, with a higher percentage contribution than FLAVA and AC in the first two factors.

Samples from the two years of ‘Sousón’ (SO-18 and SO-19), ‘Castañal’ (CS-18 and CS-19), ‘Caiño Bravo’ (CB-18 and CB-19), ‘Caiño Longo 1’ and ‘Caiño Longo 2’ (CL1-18, CL1-19 and CL2-18, CL2-19), ‘Ferrón’ (FE-18 and FE-19), and ‘Espadeiro’ (ES-18 and ES-19), and, from 2019, for ‘Caiño Tinto’ (CT-19), were located in the negative side of F1 and most of them in the positive side of F2 (first quadrant). These varieties that had low % of non-anthocyanin families were included in the same reconstructed population (RPP), RPP1a, by Díaz-Losada et al. [72], and other varieties such as ‘Albarín Tinto’ (AT-18 and AT-19), ‘Mencía’ (ME-18 and ME-19), ‘Pedral’ (PE-19), or ‘Merenzao’ (MZ-19), belong to RPP1b. Both RPP correspond to northwestern Iberian Peninsula varieties. In both years, ‘Corbillón’ (CO-18 and CO-19), ‘Evega 3’ (EV3-18 and EV3-19), ‘Evega 6’ (EV6-18 and EV6-19), and ‘Xafardán’ (XA-18 and XA-19) were grouped in the positive side of F1 and negative of F2. These varieties were characterized by high values of % FLAVA; % FLAVO values allowed the differentiation of samples of ‘Brancellao’ (BR-18 and BR-19), ‘Evega 4’ (EV4-18 and EV4-19), ‘Garnacha’ (GA-18 and GA-19), ‘Híbrido’ (HI-18 and HI-19), ‘Merenzao’ (MZ-18 and MZ-19), and ‘Picapoll Negro’ (PN-18 and PN-19); within this last group, a subgroup formed by ‘Brancellao’ (BR-18 and BR-19), ‘Evega 4’ (EV4-18 and EV4-19), ‘Garnacha’ (GA-18 and GA-19′), and ‘Picapoll Negro’ (PN-18 and PN-19) could be distinguished by their high % AC values.

The following PCAs were focused on determining the capacity of the different compounds belonging to FLAVA, FLAVO, and HAC and HBA families to discriminate cultivars. In these cases, the percentages of each compound (or the same aglycone) with respect to the respective NAN values were used.

For FLAVO compounds, the percentages of % Iso, Kaemp, Quer, and My with respect to the NAN from the 2018 and 2019 were used. Figure 6 reflects the plot with the distribution of the variables and samples obtained.

F1 and F2 explained 84.23% of the total variance (67.87% and 26.87%, respectively). From the PCA results, different FLAVO compounds contribute to F1 in percentages of 39, 37, 18, and a 6 by % Quer, % My, % Kaemp, and % Iso, respectively, while they do it to F2 in percentages of 68, 23, 5, and 4 by % Iso, % Kaemp, % My, and % Quer, respectively. Three groups can be distinguished: the first one includes those varieties characterized by their % Kaemp in 2018 and 2019, including samples from ‘Gran Negro’ (GN-18 and GN-19), ‘Mandón’ (MA-18 and MA-19), ‘Mencía’ (ME-18 and ME-19), and ‘Caiño Longo 2’ (CL2-18 and CL2-19); the second one includes samples with high % My which includes ‘Sousón’ (SO-18 and SO-19), ‘Ferrón’ (FE-18 and FE-19), ‘Castañal’ (CS-18 and CS-19), ‘Espadeiro’ (ES-18 and ES-19), ‘Caiño Bravo’ (CB-18 and CB-19), ‘Caiño Longo 1’ (CL1-18 and CL1-19), ‘Caiño Tinto’ (CT-18 and CT-19), and ‘Pan y Carne’ (PC-18 and PC-19); the third group includes those varieties with high % Quer in both year: ‘Moscatel de Hamburgo’ (MH-18 and MH-19), ‘Brancellao’ (BR-18 and BR-19), and ‘Xafardán’ (XA-18 and XA-19). It is noted that, in general, for a given variety, samples from 2018 are located close to those of 2019 and a clear interannual differentiation did not exist. When Ferrandino et al. [13] tested the flavonol profile to discriminate 34 red *V. vinifera* cultivars, they found that Que and My efficiently discriminated varieties.

The resulting plot of PCA carried out with FLAVA compounds percentages (% Cat, Epi, Galo, GaloEpi, PB1 and PB2) with respect to the NAN from the years 2018 and 2019 is shown in Figure 7. In this case, F1 and F2 explained 65.20% of total variance (39.29% and 25.91%, respectively). F1 is mainly defined by % Cat and % Epi in the positive side and % Galo in the negative side. A clear ‘year’ effect, with most 2018 samples in both positive sides of F1 and F2 and most 2019 samples in both negative sides of these axes, was observed. In this sense, Rienth et al. [73] suggest that temperature impacts tannin synthesis and galloylation in the young berry. According to these studies, the tannin synthesis and galloylation are either impaired by high temperature during the first phase of berry growth or cool day temperature stimulates tannin synthesis. Therefore, the meteorological conditions of 2018, a slightly warmer and rainier season than 2019, could explain these results.

The last PCA shown in Figure 8 was carried out with phenolic acids compounds percentages (% COU, GA, CF, and FR) with respect to the NAN. F1 and F2 accounted for 74.09% of total variance (44.06 and 30.03, respectively). As Figure 8 reflects, % GA is correlated with F1 in accordance with PCA results, which contribute to this factor by 56%, while % COU contribute at 67% with respect to F2. ‘Corbillón’ (CO-18 and CO-19), ‘Evega 3’ (EV3-18 and EV3-19), ‘Evega 6’ (EV6-18 and EV6-19), ‘Moscatel de Hamburgo’ (MH-18 and MH-19), ‘Merenzao’ (MZ-18 and MZ-19), ‘Xafardán’ (XA-18 and XA-19), and ‘Zamarrica’ (ZA-18 and ZA-19) are separated from the rest of samples.

It is noticed that, especially in the non-anthocyanin profile’s PCA (Figure 5), as well as in the flavonols, PCA (Figure 6) could be understood as a varietal aggrupation of varieties belonging to the same genetic group RRP1 defined by Díaz-Losada et al. [72], which would imply certain chemotaxonomic effects of the non-anthocyanin phenolic compounds’ profiles, especially those of the FLAVO compounds. It can be seen that the yearly effect appeared only in FLAVA (Figure 7).

## 3. Materials and Methods

### 3.1. Plant Material and Environmental Conditions

Twenty-eight genotypes of *Vitis vinifera* L. red grapes from ‘Estación de Viticultura y Enología de Galicia’ (EVEGA) germplasm bank were analyzed in 2018 and 2019 vintages. The vineyard is situated in Leiro, Ourense, Galicia (42°21′34.5″ N 8°07′08.2″ W, elevation 87 m), in the Eurosiberian biogeographic region, with a marked Mediterranean influence. Vines are grafted on 196-17C rootstock, trained to a vertical trellis system (VSP) on Cordon Royat and spaced 1.2 × 1.8 m. Their average age is 30 years old, and every cultivar is present in duplicate plots of 6 to 11 vines. The site has an adamellitic granite soil of two micas (IGME). The germplasm bank has a surface area of 8600 m^2^ and an east–west orientation. Meteorological conditions (Maximum, minimum, mean temperatures, and rainfall) were registered by an automatic meteorological station (iMETOS, Pessl Instruments GmbH, Weiz, Austria) located in the vineyard. Two climate indices related to thermal conditions were calculated: Heliothermal index (HI) [74] and Cool night Index (CI) [34]. 

### 3.2. Maturation State Assessment

In both vintages, the probable alcoholic degree (PAD) of grapes was monitored weekly from veraison to harvest. The goal was to harvest the grapes at 12.5–13.5° PAD (usual criteria for red wines of this region) and according to their health status. As is reflected the Table 1, it was possible for most of the samples, with some exceptions. Six hundred grapes were collected from the bottom, central, and top part of the bunch from both plots of each variety in a proportional way depending on the number of vines of each one. From those samples, 200 grapes were obtained, weighed (analytical balance, Mettler Toledo PL602-S, Columbus, OH, USA), and frozen at −20 °C until their polyphenolic extraction and analysis procedure. Two aliquots of 100 grapes were used for the technological maturity parameters: probable alcoholic degree (PAD) and total soluble solids (TSS, °Brix), and titratable acidity (TA, g tartaric acid L^−1^) was determined according to official methods [75].

### 3.3. Anthocyanin and Non-Anthocyanin Polyphenolic Compounds Acquirement

#### 3.3.1. Chemicals

Acetonitrile, formic acid, and methanol (MeOH) were of HPLC-gradient grade and were purchased from Panreac (Barcelona, Spain). The water was treated in a Milli-Q system (Millipore, Milford, MA, USA). Catechin, epicatechin, catechin gallate, epicatechin gallate, procyanidins B1 and B2, myricetin-3-glucoside, myricetin-3-galactoside, querce-tin-3-glucoside, quercetin-3-glucuronide, kaempherol-3-glucoside, kaempherol-3-rutinoside, isorhamnetin-3-glucoside, and isorhamnetin-3-rutinoside were purchased by Extrasynthese (Lyon, France) and gallic, caffeic, ferulic, and coutaric acids were purchased from Aldrich (Munich, Germany).

#### 3.3.2. Extraction of Polyphenols from Grapes

Polyphenolic substances of approximately 50 g of healthy, frozen whole grapes were extracted with 50 mL methanol/water/formic acid (50:48.5:1.5, *v*/*v*/*v*/*v*) according to the method described by Portu et al. [76] with slight modifications. The methanol acid mixture was added to the frozen gapes and homogenized (Moulinex 180 W grinder, Alençon, France), sonicated for 10 min at 50 Hz (Grant XUB5, Cambridge, England), and centrifuged at 5000 rpm for 10 min (Allegra 25R, Beckman Coulter, DE, USA). The supernatant was separated, and the resulting pellet was extracted up to three times. Supernatants obtained were combined in a flask and the volume was brought to 200 mL with the extraction mixture and stored at −20 °C until analysis. Three extractions were performed for each sample of a given cultivar. 

For the analysis of ANs, the extract was injected directly into the HPLC.

Isolation of non-anthocyanin compounds was carried out based on previous works [30]. First, grape phenolic extracts (3 mL) were diluted with 9 mL of 0.1 N HCl and passed through PCX SPE cartridges, (Chromafil PET 20/25, Machery-Nagel, Düren, Germany). The non-anthocyanin phenolic substances were eluted with 3 × 5 mL of methanol. 

The eluate containing FLAVO, FLAVA, HCA, and HBA was separately dried in a rotary evaporator (40 °C) and redissolved in 1.5 mL of 20% (*v*/*v*) methanol aqueous solution.

#### 3.3.3. Identification and Quantification

Identification and quantification of phenolic compounds were performed by HPLC analysis. An Agilent 1200 LC system (Agilent Technologies, Palo Alto, CA, USA), equipped with a degasser, quaternary pump, column oven, 1290 infinity autosampler, UV-Vis diode-array detector (DAD), fluorescence spectrophotometer detector (FLD), and the Chemstation software package for LC 3D systems (Agilent Technologies, Palo Alto, CA, USA) to control the instrument and for data acquisition and analysis, were used. Separation was performed in a Licrospher^®^ (Darmstadt, Germany) 100 RP-18 reversed-phase column (250 × 4.0 mm; 5 µm packing; Agilent Technologies, Palo Alto, CA, USA) with pre-column Licro-spher^®^ 100 RP-18 (4 × 4 mm; 5 µm packing; Agilent Technologies, Palo Alto, CA, USA). We injected 20 μL of each sample in triplicate and chromatographic conditions were based on those described by Castillo-Muñoz et al. [77]. The following eluents and solvents gradient were used: (A) acetonitrile/water/formic acid, (3:88.5:8.5, *v*/*v*/*v*), (B) acetonitrile/water/formic acid (50:41.5:8.5, *v*/*v*/*v*), and (C) methanol/water/formic acid (90:1.5:8.5, *v*/*v*/*v*), maintaining the column at 40 °C and the flow rate in 0.63 mL·min^−1^. The linear solvents gradient was as follows: 0 min, 96% A, and 4% B; 7 min, 96% A, and 4% B; 38 min, 70% A, 17% B, and 13% C; 52 min, 50% A, 30% B, and 20% C; 52.5 min, 30% A, 40% B, and 30% C; 57 min, 50% B, and 50% C; 58 min, 50% B, and 50% C; 65 min, 96% A, and 4% B.

Absorbances at 320, 360, and 520 nm were measured by the DAD detector to quantify phenolics acids, flavonols, and anthocyanin. Excitation at 280 and emission at 320 nm were measured by the FLD detector to quantify the flavanols compounds.

ANs present in extracts were identified and quantified as simple glucosides-nonacylated, acetyl derivatives, and coumaroyl-derivative forms of delphinidin, cyanidin, petunidin, peonidin, and malvidin. The total amount of AN was given in mg of malvidine-3-glucoside·kg^−1^ fresh berry weight (FW). FLAVA identified were: (+)-catechin, (−)-epicatechin, gallocatechin, epigallocatechin, and the procyanidins B1 and B2. These were quantified as mg of (+)-catechin·kg^−1^ FW. FLAVO identified were myricetin, kaempherol, quercetin, isorhamnetin, and the 3-glucosides and 3-rutinoside of quercetin and kaempherol, quercetin-3-galactoside and quercetin-3-glucuronide. These were quantified as mg of quercetin-3-glucoside·kg^−1^ FW. Finally, gallic, caffeic, ferulic, and coutaric acid were identified and quantified as mg of caffeic acid·kg^−1^ FW.

Each result is the mean value of 6 determinations (2 HPLC/extract × 3 extracts/sample).

### 3.4. Data Analysis

Sample analyses were performed with XLSTAT-Pro 201610 (Addinsoft 2009, Paris, France) statistical software package. With the aim of investigating the influences of ‘variety’, ‘year’, and their interaction ‘variety × year’ on each parameter evaluated, data were subjected to analysis of variance (ANOVA), selecting *p* ≤ 0.001, *p* ≤ 0.01, and *p* ≤ 0.05 for significance of comparisons. The interaction between varieties and years was evaluated by calculating the least-squares means (LS means) selecting *p* ≤ 0.001, *p* ≤ 0.01, and *p* ≤ 0.05 for significance of comparisons. Significance of differences between each year was based on Student’s *t*-test. The data of non-anthocyanin compounds profiles were submitted to principal component analysis (PCA) with the aim of differentiating varieties based on the studied variables association.

## 4. Conclusions

In this study, flavanol, flavonol, and phenolic acids profiles (non-anthocyanin phenolic compounds NAN) of 28 genotypes of *Vitis vinifera* L. grapes from the EVEGA germplasm bank were defined in two consecutive vintages. To the best of our knowledge, for almost all the varieties under study, this is the first time that this characterization is reported. The percentage of total NAN with respect to the total polyphenol content (TPC) values was calculated for each sample and established into three categories: those high percentage NAN varieties (NANV), those varieties showing low percentage of NAN (ANV), and finally those varieties showing medium percentages of NAN (NANAV). ‘Xafardán’ and ‘Zamarrica’, classified as NANAV, had high values of TPC and showed balanced percentages of anthocyanins and NAN compounds that can potentially act as copigments. On the other hand, some NANV, such as ‘Corbillón’, ‘Evega 3’, and ‘Evega 6’, could be considered in terms of their flavanols and gallic acid in plurivarietal winemaking, a possible line of future research being the analysis of these varieties in terms of whether their color stability complies with the results obtained in this investigation.

These results could help in the selection of varieties to be disseminated and in the identification of the most appropriate agronomic and oenological techniques that should be performed on them. Additionally, this work provides information about the differentiating capacity of phenolic compounds and their use as varietal markers and chemotaxonomic tools. Flavonol compounds (myricetin, quercetin, and kaempherol) were found to be useful compounds in the differentiation the different NANV samples.

## Figures and Tables

**Figure 1 plants-12-00004-f001:**
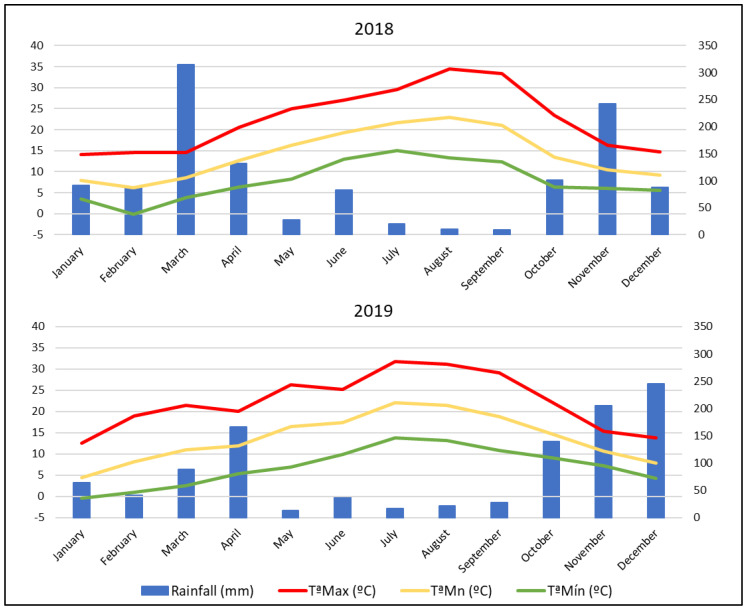
Climatic conditions in 2018 and 2019 vintages. Max T, Mean T, and Min T are average values of the maximum, mean, and minimum temperatures, respectively.

**Figure 2 plants-12-00004-f002:**
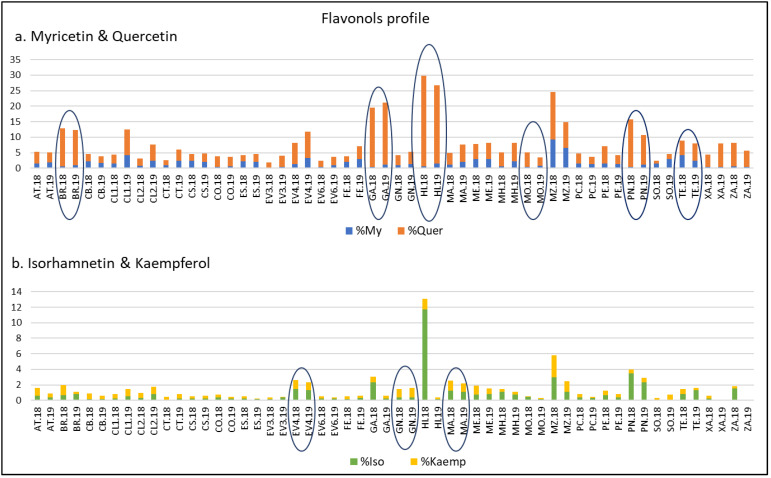
Flavonols profile with respect to the total polyphenolic content of varieties from the EVEGA germplasm bank in 2018 and 2019 vintages. % MY: percentage of myricetin compounds; % Quer: percentage of quercetin compounds; % Iso: percentage of isorhamnetin compounds; % Kaemp: percentage of kaempherol compounds. See Table 1 for varieties abbreviations.

**Figure 3 plants-12-00004-f003:**
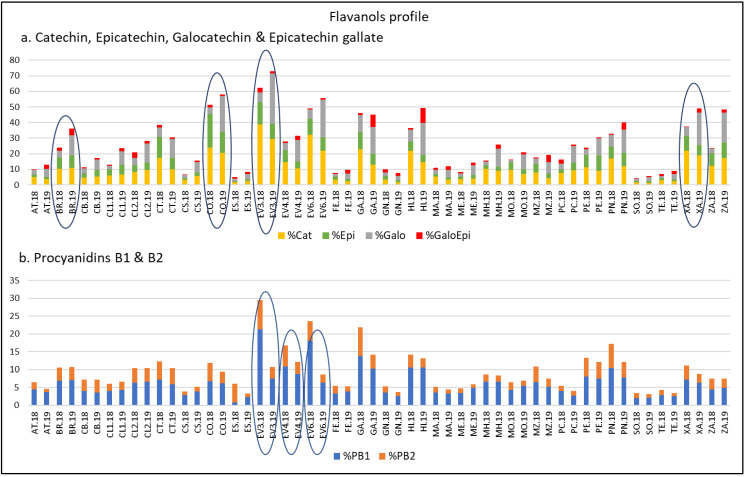
Flavanols profile of varieties from the EVEGA germplasm bank in 2018 and 2019 vintages. % Cat, % Galo, % Epi, % GaloEpi, % PB1, and % PB2 refer to percentage of catechin, gallocatechin, epicatechin, epicatechin gallate, procyanidin B1, and procyanidin B2, respectively, with respect to TPC values. See Table 1 for varieties abbreviations.

**Figure 4 plants-12-00004-f004:**
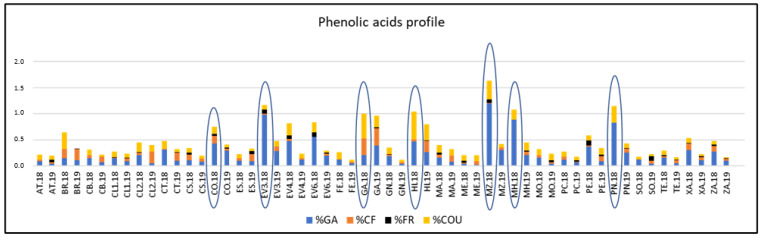
Phenolic acids profile respect to the total polyphenol content of varieties from the EVEGA germplasm bank in 2018 and 2019 vintages. % CF; % COU; % GA and % FR are percentages of caffeic acid, coutaric acid, gallic acid and ferulic acid, respectively. See Table 1 for varieties abbreviations.

**Figure 5 plants-12-00004-f005:**
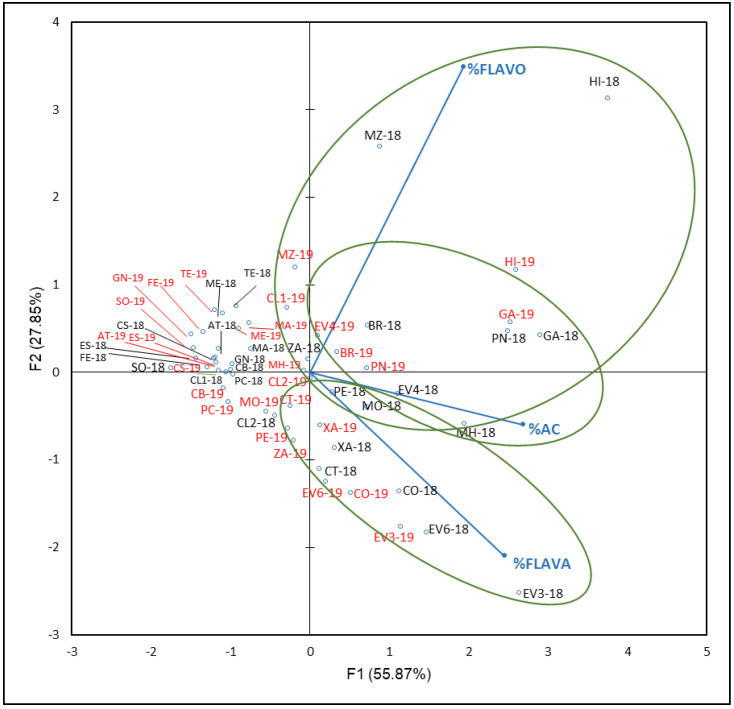
Principal component analysis on non-anthocyanin phenolic compounds profile (percentage) of grapes from the EVEGA germplasm bank in 2018 and 2019 vintages. % FLAVO, % FLAVA, and % AC are percentages of flavonols, flavanols and phenolic acids, respectively. Data for the black and red varieties reflect the data for 2018 and 2019 harvests respectively. See Table 1 for varieties abbreviations.

**Figure 6 plants-12-00004-f006:**
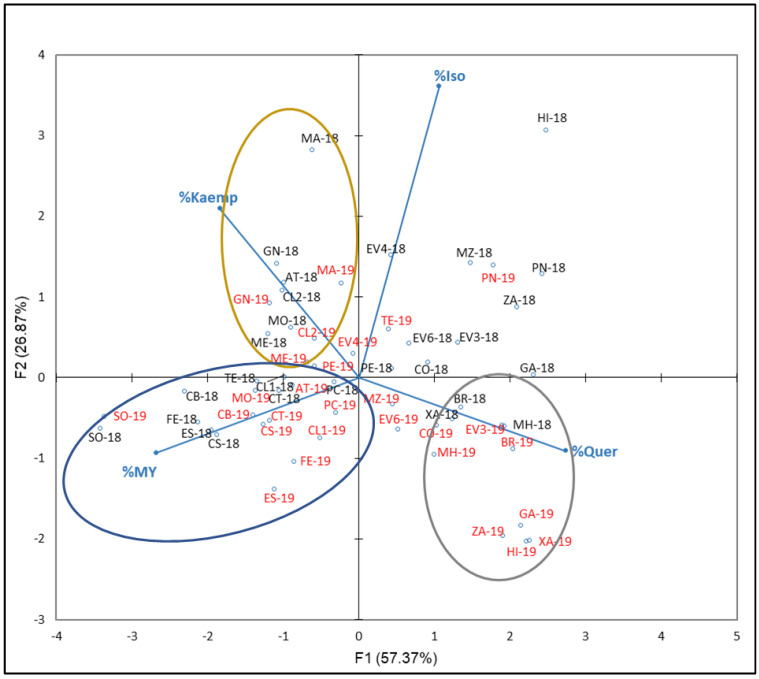
Principal component analysis on flavonol family compounds profile (percentage) of grapes from the EVEGA germplasm bank in 2018 and 2019 vintages. % MY, % Kaemp, % Iso, and % Quer are percentages of myricetin, kaempherol, isorhamnetin, and summatory of quercetin compounds, respectively. Data for the black and red varieties reflect the data for 2018 and 2019 harvests respec-tively. See Table 1 for varieties abbreviations.

**Figure 7 plants-12-00004-f007:**
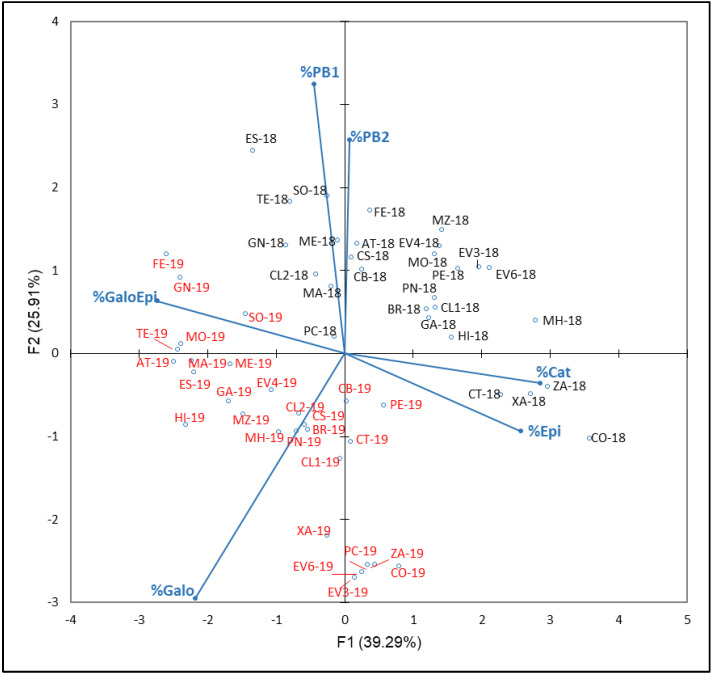
Principal component analysis on flavanol family compounds profile (percentage) of grapes from the EVEGA germplasm bank in 2018 and 2019 vintages. % Cat, % Galo, % Epi, % GaloEpi, % PB1 and % PB2 are percentages of catechin, gallocatechin, epicatechin, epicatechin gallate, procyanidin B1 and procyanidin B2, respectively, with respect to TPC values. Data for the black and red varieties reflect the data for 2018 and 2019 harvests respectively. See Table 1 for varieties abbreviations.

**Figure 8 plants-12-00004-f008:**
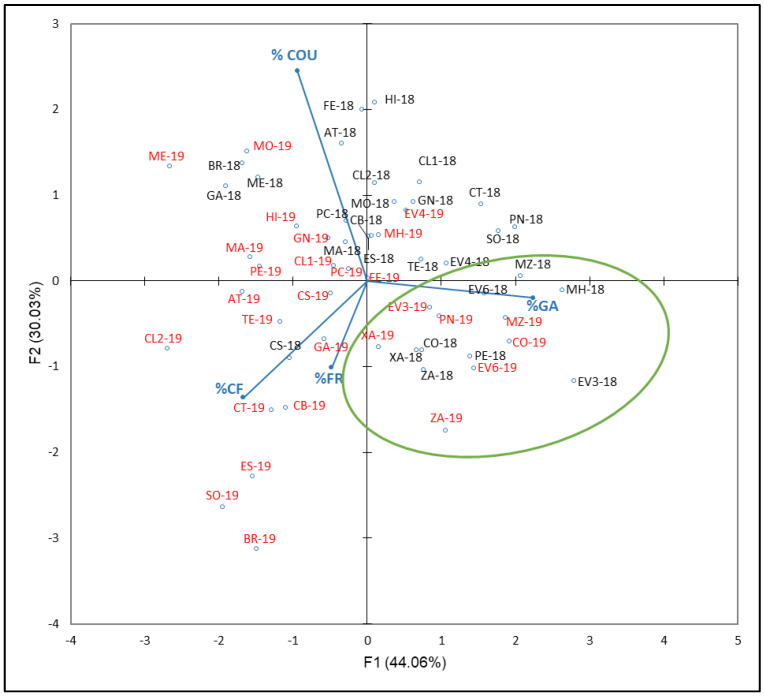
Principal component analysis on phenolic acids compounds profile (percentage) of grapes from the EVEGA germplasm bank in 2018 and 2019 vintages. % CF, % COU, % GA, and % FR are percentages of caffeic acid, coutaric acid, gallic acid, and ferulic acid, respectively. Data for the black and red varieties reflect the data for 2018 and 2019 harvests respectively. See Table 1 for varieties abbreviations.

**Table 1 plants-12-00004-t001:** Data of harvest, berry weight, and composition of grapes from the EVEGA germplasm bank in 2018 and 2019 vintages.

Variety	Abbreviation	Grape Type	Harvest Data	Berry Weight (g)	PAD	Technological Maturity
**‘Albarín Tinto’**	**AT**	**W**	21 September 2018	1.93 ± 0.00	13.24 ± 0.22	4.93
			10 September 2019	1.55 ± 0.03	14.4 ± 0.10	4.14
**‘Brancellao’**	**BR**	**W**	21 September 2018	1.70 ± 0.04	13.31 ± 0.11	4.00
			10 September 2019	1.71 ± 0.03	13.1 ± 0.09	3.36
**‘Caíño Bravo’**	**CB**	**W**	27 September 2018	1.73 ± 0.17	12.83 ± 0.21	3.75
			17 September 2019	1.62 ± 0.09	12.6 ± 0.13	2.91
**Caíño Longo 1’**	**CL1**	**W**	17 September 2018	1.90 ± 0.05	13.24 ± 0.15	3.34
			17 September 2019	1.84 ± 0.15	13.3 ± 0.10	2.56
**‘Caíño Longo 2’**	**CL2**	**W**	21 September 2018	2.00 ± 0.06	13.51 ± 0.17	3.55
			13 September 2019	1.98 ± 0.09	13.9 ± 0.13	2.69
**‘Caíño Tinto’**	**CT**	**W**	27 September 2018	2.47 ± 0.09	11.47 ± 0.18	2.18
			17 September 2019	2.43 ± 0.18	11.9 ± 0.08	2.51
**‘Castañal’**	**CS**	**W**	27 September 2018	2.10 ± 0.07	12.97 ± 0.22	6.03
			17 September 2019	1.92 ± 0.00	12.5 ± 0.10	2.81
**‘Corbillón’**	**CO**	**W**	17 September 2018	2.03 ± 0.13	13.31 ± 0.18	4.47
			2 September 2019	1.38 ± 0.12	13.17 ± 0.20	3.10
**‘Espadeiro’**	**ES**	**W**	27 September 2018	2.37 ± 0.16	13.17 ± 0.14	4.52
			17 September 2019	1.57 ± 0.09	12.8 ± 0.11	3.81
**‘Evega 3’**	**EV3**	**W**	12 September 2018	2.07 ± 0.01	14.55 ± 0.03	4.39
			2 September 2019	1.55 ± 0.10	13.58 ± 0.10	3.09
**‘Evega 4’**	**EV4**	**W**	21 September 2018	2.57 ± 0.02	13.03 ± 0.11	4.57
			17 September 2019	2.65 ± 0.26	14 ± 0.16	5.67
**‘Evega 6’**	**EV6**	**W**	17 September 2018	2.53 ± 0.01	13.72 ± 0.21	4.59
			10 September 2019	1.72 ± 0.00	13.6 ± 0.19	3.82
**‘Ferrón’**	**FE**	**W**	21 September 2018	2.13 ± 0.13	13.24 ± 0.14	4.83
			13 September 2019	1.67 ± 0.09	13.5 ± 0.18	2.22
**‘Garnacha’**	**GA**	**W**	13 September 2018	2.20 ± 0.22	13.44 ± 0.05	4.18
			2 September 2019	1.41 ± 0.08	13.24 ± 0.08	3.15
**‘Gran Negro’**	**GN**	**W**	27 September 2018	2.70 ± 0.11	10.8 ± 0.20	4.90
			17 September 2019	2.03 ± 0.05	10.7 ± 0.06	3.39
**‘Híbrido’**	**HI**	**-**	12 September 2018	3.50 ± 0.86	13.44 ± 0.12	4.69
			4 September 2019	2.52 ± 0.21	12.2 ± 0.18	4.14
**‘Mandón’**	**MA**	**W**	17 September 2018	2.00 ± 0.11	13.24 ± 0.15	5.04
			13 September 2019	1.67 ± 0.11	13.1 ± 0.09	3.95
**‘Mencía’**	**ME**	**W**	13 September 2018	2.10 ± 0.00	12.62 ± 0.02	5.59
			2 September 2019	1.44 ± 0.05	12.76 ± 0.11	4.49
**‘Merenzao’**	**MZ**	**W**	12 September 2018	2.07 ± 0.07	14.89 ± 0.14	7.17
			2 September 2019	1.98 ± 0.03	13 ± 0.22	3.66
**‘Moscatel de Hamburgo’**	**MH**	**T**	12 September 2018	5.07 ± 0.74	13.1 ± 0.08	5.36
			4 September 2019	3.76 ± 0.17	12.7 ± 0.17	3.48
**‘Mouratón’**	**MO**	**W**	17 September 2018	3.10 ± 0.01	13.92 ± 0.12	5.04
			4 September 2019	2.90 ± 0.20	12.9 ± 0.13	3.64
**‘Pan y Carne’**	**PC**	**W**	13 September 2018	2.35 ± 0.01	14.55 ± 0.21	4.64
			2 September 2019	2.06 ± 0.10	14.1 ± 0.15	2.79
**‘Pedral’**	**PE**	**W**	21 September 2018	1.93 ± 0.10	13.58 ± 0.08	3.52
			19 September 2019	1.79 ± 0.06	13.3 ± 0.14	3.04
**‘Picapoll Negro’**	**PN**	**W**	27 September 2018	3.13 ± 0.28	13.31 ± 0.09	3.80
			13 September 2019	2.73 ± 0.09	13.4 ± 0.10	3.32
**‘Sousón’**	**SO**	**W**	27 September 2018	2.37 ± 0.08	12.97 ± 0.14	3.60
			13 September 2019	1.60 ± 0.03	13 ± 0.05	3.25
**‘Tempranillo’**	**TE**	**W**	13 September 2018	2.63 ± 0.19	13.92 ± 0.15	5.27
			2 September 2019	1.55 ± 0.10	13 ± 0.07	3.86
**‘Xafardán’**	**XA**	**W**	28 September 2018	1.07 ± 0.00	15.38 ± 0.16	2.90
			13 September 2019	0.73 ± 0.01	15.3 ± 0.04	2.95
**‘Zamarrica’**	**ZA**	**W**	25 September 2018	1.27 ± 0.11	14.48 ± 0.18	5.00
			13 September 2019	1.13 ± 0.03	12.8 ± 0.12	2.40

PAD: Probable Alcoholic Degree (% *v*/*v*); Technological maturity: ratio TSS (°Brix) and TA (g tartaric acid·L^−1^).

**Table 2 plants-12-00004-t002:** Non-anthocyanin, anthocyanin, and total polyphenolic content. (Values for phenolic compounds of varieties from the EVEGA germplasm bank in 2018 and 2019 vintages are expressed as mg·kg^−1^ fresh berry weight (FW) for NAN, AN and TPC values and percentages of non-anthocyanin and anthocyanin were used in % NAN and % AN, respectively).

Variety	Year	NAN	%NAN	AN	%AN	TPC
**AT**	2018	206.85 ± 4.47	23.45 ± 0.50	675.07 ± 9.72	76.55 ± 0.50	881.92 ± 0.40
	2019	339.47 ± 57.70	23.75 ± 2.96	**1089.76 *** ± 38.97	76.25 ± 2.96	**1429.23 *** ± 76.72
**BR**	2018	318.55 ± 26.83	49.87 ± 1.83	320.20 ± 15.55	50.13 ± 1.83	638.75 ± 30.33
	2019	**532.81 *** ± 33.21	**60.66 *** ± 2.08	**345.57 *** ± 8.50	**39.34 *** ± 2.08	**878.38 *** ± 24.75
**CB**	2018	353.15 ± 13.50	24.17 ± 0.55	1107.72 ± 75.76	75.83 ± 0.55	1460.87 ± 88.98
	2019	**441.63 *** ± 21.42	**29.08 *** ± 1.12	1076.99 ± 64.04	**70.92 *** ± 1.12	1518.61 ± 76.63
**CL1**	2018	289.33 ± 84.69	24.45 ± 4.62	894.01 ± 53.17	75.55 ± 4.62	1183.34 ± 124.11
	2019	**548.96 *** ± 59.23	38.23 * ± 2.89	887.14 ± 54.70	**61.77 *** ± 2.89	1436.10 ± 79.39
**CL2**	2018	313.53 ± 35.95	38.94 ± 1.18	491.56 ± 31.84	61.06 ± 1.18	805.09 ± 67.93
	2019	**570.77 *** ± 33.35	**48.79 *** ± 1.51	**599.07 *** ± 13.78	**51.21 *** ± 1.51	**1169.84 *** ± 37.20
**CT**	2018	566.31 ± 85.81	54.30 ± 4.39	476.65 ± 12.08	45.70 ± 4.39	1042.96 ± 73.90
	2019	465.93 ± 45.11	48.28 ± 1.86	499.13 ± 17.59	51.72 ± 1.86	965.06 ± 59.69
**CS**	2018	211.85 ± 35.51	15.68 ± 1.38	1139.35 ± 68.34	84.32 ± 1.38	1351.20 ± 107.68
	2019	**617.20 *** ± 87.02	**26.59 *** ± 1.80	**1703.77 *** ± 87.88	**73.41 *** ± 1.80	**2320.98 *** ± 174.80
**CO**	2018	936.92 ± 45.32	68.44 ± 0.53	432.09 ± 13.11	31.56 ± 0.53	1369.01 ± 55.57
	2019	**1483.77 *** ± 52.44	71.96 ± 1.57	**578.27 *** ± 60.73	28.04 ± 1.57	**2062.04 *** ± 107.41
**ES**	2018	201.73 ± 20.06	15.85 ± 1.26	1071.17 ± 18.39	84.15 ± 1.26	1272.90 ± 25.29
	2019	**281.80 *** ± 19.71	16.96 ± 1.11	**1379.45 *** ± 25.79	83.04 ± 1.11	**1661.25 *** ± 25.73
**EV3**	2018	1066.50 ± 20.99	95.31 ± 0.82	52.44 ± 8.04	4.69 ± 0.82	1118.94 ± 12.41
	2019	1154.68 ± 87.12	**88.56 *** ± 1.83	**149.20 *** ± 16.03	**11.44 *** ± 1.83	**1303.89 *** ± 74.62
**EV4**	2018	361.78 ± 56.13	56.29 ± 4.91	280.93 ± 27.09	43.71 ± 4.91	642.72 ± 43.74
	2019	604.59 ± 94.81	57.94 ± 5.69	**438.84 *** ± 34.10	42.06 ± 5.69	**1043.43 *** ± 69.92
**EV6**	2018	892.53 ± 74.21	76.51 ± 2.08	273.96 ± 10.19	23.49 ± 2.08	1166.49 ± 65.39
	2019	**1127.73 *** ± 63.11	**68.80 *** ± 1.53	511.37 * ± 8.52	**31.20 *** ± 1.53	**1639.10 *** ± 55.93
**FE**	2018	295.46 ± 41.11	17.94 ± 1.29	1351.84 ± 87.98	82.06 ± 1.29	1647.30 ± 110.95
	2019	372.69 ± 54.22	22.76 ± 1.82	1264.70 ± 50.55	77.24 ± 1.82	1637.39 ± 104.15
**GA**	2018	182.60 ± 6.63	91.37 ± 0.18	17.24 ± 0.86	8.63 ± 0.18	199.84 ± 6.87
	2019	271.59 ± 41.65	**82.31 *** ± 0.95	**58.38 *** ± 4.87	**17.69 *** ± 0.95	**329.98 *** ± 46.52
**GN**	2018	165.65 ± 18.27	21.34 ± 1.58	610.53 ± 10.07	78.66 ± 1.58	776.18 ± 28.33
	2019	209.70 ± 26.77	18.45 ± 1.01	**927.17 *** ± 59.39	81.55 ± 1.01	**1136.88 *** ± 85.89
**HI**	2018	201.78 ± 9.92	94.58 ± 1.82	11.56 ± 3.76	5.42 ± 1.82	213.34 ± 6.37
	2019	**381.36 *** ± 42.24	90.55 ± 1.46	**39.81 *** ± 6.04	9.45 ± 1.46	**421.17 *** ± 44.35
**MA**	2018	190.58 ± 32.61	24.22 ± 2.56	596.33 ± 26.82	75.78 ± 2.56	786.91 ± 51.61
	2019	266.35 ± 27.51	26.73 ± 2.86	730.28 ± 61.89	73.27 ± 2.86	**996.63 *** ± 61.65
**ME**	2018	215.07 ± 12.59	22.89 ± 1.16	724.66 ± 23.20	77.11 ± 1.16	939.74 ± 7.56
	2019	**314.06 *** ± 7.99	**30.32 *** ± 1.21	719.92 ± 17.26	69.50 * ± 1.21	**1035.80 *** ± 6.87
**MZ**	2018	194.52 ± 72.24	57.31 ± 8.36	144.92 ± 5.66	42.69 ± 8.36	339.44 ± 77.83
	2019	353.31 ± 114.94	50.47 ± 10.51	**346.66 *** ± 24.34	49.53 ± 10.51	**699.96 *** ± 91.81
**MH**	2018	280.17 ± 101.14	45.71 ± 7.80	332.77 ± 16.52	54.29 ± 7.80	612.94 ± 118.62
	2019	390.16 ± 44.06	48.68 ± 3.80	**411.26 *** ± 21.04	51.32 ± 3.80	801.42 ± 35.64
**MO**	2018	203.38 ± 27.90	21.25 ± 1.79	753.53 ± 22.71	78.75 ± 1.79	956.90 ± 50.68
	2019	**310.30 *** ± 8.17	**25.65 *** ± 0.37	**899.44 *** ± 11.83	**74.35 *** ± 0.37	**1209.74 *** ± 18.51
**PC**	2018	296.33 ± 43.19	27.52 ± 1.69	780.32 ± 70.56	72.48 ± 1.69	1076.65 ± 90.97
	2019	**509.00 *** ± 32.62	**34.33 *** ± 1.51	**973.65 *** ± 15.89	**65.67 *** ± 1.51	**1482.66 *** ± 35.71
**PE**	2018	435.12 ± 28.35	45.96 ± 4.40	511.59 ± 71.48	54.04 ± 4.40	946.71 ± 28.89
	2019	633.31 * ± 37.23	48.22 ± 1.34	680.09 * ± 24.71	51.78 ± 1.34	**1313.40 *** ± 53.51
**PN**	2018	230.96 ± 34.37	70.91 ± 2.07	94.76 ± 6.38	29.09 ± 2.07	325.71 ± 39.04
	2019	387.29 ± 66.96	66.29 ± 3.73	**196.91 *** ± 13.56	33.71 ± 3.73	**584.20 *** ± 70.17
**SO**	2018	189.10 ± 20.46	10.33 ± 1.75	1640.77 ± 129.84	89.67 ± 1.75	1829.87 ± 110.87
	2019	**364.70 *** ± 21.15	14.15 ± 1.21	**2213.46 *** ± 100.20	85.85 ± 1.21	**2578.16 *** ± 82.82
**TE**	2018	217.17 ± 69.69	21.95 ± 4.78	772.05 ± 40.92	78.05 ± 4.78	989.22 ± 103.37
	2019	323.46 ± 7.20	22.26 ± 0.58	1129.59 * ± 56.07	77.74 ± 0.58	**1453.05 *** ± 62.09
**XA**	2018	641.29 ± 3.97	54.23 ± 1.45	541.28 ± 24.88	45.77 ± 1.45	1182.58 ± 24.20
	2019	**1167.56 *** ± 35.53	**66.02 *** ± 0.65	**600.97 *** ± 3.20	**33.98 *** ± 0.65	**1768.53 *** ± 37.02
**ZA**	2018	475.00 ± 131.40	41.13 ± 6.40	679.89 ± 10.38	58.87 ± 6.40	1154.90 ± 141.19
	2019	**1186.04 *** ± 10.11	**61.64 *** ± 0.45	**737.96 *** ± 15.95	**38.36 *** ± 0.45	**1923.99 *** ± 22.64
**Variety**		***	***	***	***	***
**% Variation**		82.64	96.44	93.68	96.44	82.22
**Year**		***	ns	*	ns	***
**% Variation**		9.12	0.38	2.82	0.38	11.41
**Variety x Year**		***	***	***	***	***
**% Variation**		6.64	2.21	3.02	2.21	5.21

NAN: non-anthocyanin fraction; AN: anthocyanin fraction; TPC: total polyphenolic content. For ANOVA and factorial analysis: ns = non-significant; * *p* < 0.05; *** *p* < 0.001. Within each variety, * indicate significant differences among years after Student’s *t*-test. Data are reported as mean (*n* = 6). See Table 1 for varieties abbreviations.

**Table 3 plants-12-00004-t003:** Concentration of non-anthocyanin phenolic compounds: flavonols (FLAVO) from the EVEGA germplasm bank in 2018 and 2019 vintages. (Values are expressed as mg·kg^−1^ FW).

Variety	Year	My-Glc	Iso-Rut	Kaemp-Glc	Quer-Rut	Quer-Glc	Quer-GlcU
**AT**	2018	12.91 ± 1.66	5.25 ± 0.04	9.28 ± 0.13	3.25 ± 0.05	23.32 ± 0.58	6.22 ± 0.46
	2019	**25.68 *** ± 4.96	5.13 ± 2.26	7.68 ± 1.22	5.25 ± 1.20	34.81 ± 8.80	6.64 ± 0.51
**BR**	2018	3.67 ± 0.36	4.36 ± 0.22	8.14 ± 1.22	7.62 ± 1.07	68.25 ± 0.54	2.59 ± 0.05
	2019	**7.61 *** ± 0.95	7.21 ± 1.81	**2.43 *** ± 1.72	7.45 ± 1.08	88.79 * ± 8.38	**4.65 *** ± 0.37
**CB**	2018	32.37 ± 0.94	2.34 ± 0.35	10.44 ± 0.02	4.05 ± 0.12	23.62 ± 2.95	7.11 ± 0.56
	2019	24.02 ± 3.61	2.52 ± 0.67	6.32* ± 0.50	4.61 ± 0.75	24.23 ± 3.30	5.72 ± 0.57
**CL1**	2018	16.68 ± 7.73	2.96 ± 0.80	7.09 ± 2.80	3.54 ± 1.65	25.15 ± 12.73	5.92 ± 2.35
	2019	**33.07 *** ± 4.12	**4.44 *** ± 0.13	7.46 ± 1.25	5.53 ± 1.45	**54.00 *** ± 7.53	7.88 ± 1.14
**CL2**	2018	11.98 ± 4.88	4.51 ± 1.93	9.62 ± 3.34	3.56 ± 2.12	23.12 ± 5.76	6.47 ± 1.61
	2019	28.18 * ± 4.93	9.47 ± 2.68	10.85 ± 1.55	5.99 ± 1.01	**46.87 *** ± 5.49	8.76 ± 1.64
**CT**	2018	9.45 ± 0.17	1.42 ± 0.14	3.51 ± 0.03	2.93 ± 0.10	11.44 ± 1.20	3.64 ± 0.09
	2019	**22.33 *** ± 1.65	2.52 ± 0.67	5.61 ± 1.72	**4.44 *** ± 0.47	**24.98 *** ± 4.57	**5.29 *** ± 0.20
**CS**	2018	31.54 ± 4.69	3.03 ± 0.93	4.29 ± 0.19	3.14 ± 0.74	20.12 ± 4.23	5.82 ± 0.69
	2019	47.78 ± 7.31	5.77 ± 2.22	9.03 ± 2.24	7.23 ± 2.11	**48.40 *** ± 8.11	8.25 * ± 0.75
**CO**	2018	6.31 ± 0.55	4.99 ± 0.36	5.18 ± 0.06	6.67 ± 0.56	35.10 ± 3.17	4.32 ± 2.09
	2019	11.5 ± 2.99	4.84 ± 1.98	**4.01 *** ± 0.14	8.93 ± 2.88	48.75 ± 14.48	4.21 ± 0.75
**ES**	2018	28.23 ± 1.80	2.95 ± 1.26	3.57 ± 0.39	2.21 ± 0.38	17.16 ± 5.96	4.95 ± 0.12
	2019	**35.05 *** ± 1.23	2.63 ± 0.83	2.05 ± 1.30	4.20 ± 1.51	**32.08 *** ± 2.97	4.79 ± 0.14
**EV3**	2018	1.38 ± 0.33	2.33 ± 0.94	2.03 ± 1.38	2.71 ± 0.24	14.92 ± 2.65	1.5 ± 0.28
	2019	**5.27 *** ± 0.45	4.63 ± 1.62	0.88 ± 0.40	**6.31 *** ± 0.95	**39.70 *** ± 5.02	0.36 ± 0.63
**EV4**	2018	8.06 ± 1.89	9.23 ± 2.72	7.76 ± 0.05	5.63 ± 0.56	32.77 ± 9.82	5.63 ± 1.74
	2019	**34.54 *** ± 4.91	13.54 ± 3.50	**10.82 *** ± 0.35	8.19 ± 2.22	68.61 * ± 7.87	**11.06 *** ± 0.89
**EV6**	2018	4.53 ± 0.41	3.17 ± 2.47	2.77 ± 0.61	3.34 ± 0.21	17.21 ± 6.15	2.95 ± 0.64
	2019	**14.60 *** ± 0.66	4.26 ± 0.36	2.42 ± 0.19	**4.71 *** ± 0.50	**35.61 *** ± 3.61	4.25 ± 0.35
**FE**	2018	34.19 ± 9.71	3.08 ± 1.72	5.90 ± 2.80	2.53 ± 0.87	20.86 ± 12.15	6.73 ± 2.33
	2019	49.12 ± 4.67	5.66 ± 1.27	4.37 ± 0.52	**5.00 *** ± 0.41	**54.16 *** ± 6.80	8.26 ± 0.34
**GA**	2018	0.76 ± 0.28	4.67 ± 1.84	1.46 ± 0.39	3.93 ± 0.80	32.33 ± 4.38	1.82 ± 0.37
	2019	3.67 ± 1.94	0.72 * ± 0.19	1.31 ± 0.53	7.05 ± 2.59	56.74 ± 14.51	2.52 ± 0.75
**GN**	2018	7.23 ± 1.33	3.29 ± 1.60	8.24 ± 0.08	1.88 ± 0.58	18.85 ± 7.18	3.88 ± 0.38
	2019	15.25 ± 3.27	4.48 ± 1.44	13.71 ± 2.42	3.14 ± 0.60	33.98 ± 5.55	6.90 * ± 1.05
**HI**	2018	1.19 ± 0.62	25.01 ± 4.80	2.93 ± 0.16	8.68 ± 2.00	53.85 ± 14.82	0.00 ± 0.00
	2019	**6.16 *** ± 0.33	**0.71 *** ± 0.16	0.90 ± 0.95	11.11 ± 1.21	95.50 * ± 8.32	0.00 ± 0.00
**MA**	2018	8.77 ± 0.44	9.86 ± 0.26	10.13 ± 0.75	3.81 ± 0.19	21.57 ± 0.19	4.94 ± 0.11
	2019	**19.33 *** ± 2.80	11.35 ± 2.53	10.79 ± 2.01	5.34 ± 1.16	**43.66 *** ± 6.50	7.37 ± 1.50
**ME**	2018	27.00 ± 1.59	6.98 ± 0.12	10.60 ± 0.52	5.18 ± 0.59	33.27 ± 1.43	7.29 ± 0.19
	2019	30.28 ± 1.92	8.53 ± 0.95	7.53 ± 1.55	**3.74 *** ± 0.30	41.15 * ± 2.42	8.47 ± 0.67
**MZ**	2018	3.90 ± 1.68	6.88 ± 3.56	1.97 ± 0.89	4.06 ± 1.92	21.24 ± 10.48	2.28 ± 0.57
	2019	**17.21 *** ± 1.83	6.11 ± 0.88	2.71 ± 0.38	4.57 ± 0.52	39.55 ± 2.80	**4.15 *** ± 0.24
**MH**	2018	4.03 ± 0.32	4.11 ± 0.40	1.32 ± 0.06	6.45 ± 0.46	36.27 ± 0.55	2.78 ± 0.10
	2019	8.29 * ± 0.96	2.48 ± 1.67	1.25 ± 0.31	3.55 ± 1.22	29.15 ± 13.48	1.66 ± 0.51
**MO**	2018	31.14 ± 9.21	10.03 ± 5.26	9.71 ± 0.45	3.51 ± 1.46	38.10 ± 18.06	10.77 ± 2.87
	2019	46.08 ± 3.51	7.83 ± 6.25	9.55 ± 1.24	**6.92 *** ± 0.06	40.88 ± 2.20	10.07 ± 1.22
**PC**	2018	16.03 ± 1.95	4.47 ± 1.08	4.20 ± 0.46	4.58 ± 0.27	25.62 ± 2.96	4.73 ± 0.01
	2019	**20.33 *** ± 1.13	5.00 ± 2.51	1.81 ± 1.04	4.54 ± 0.69	26.60 ± 9.41	2.82 ± 1.92
**PE**	2018	13.56 ± 1.88	6.42 ± 0.75	5.80 ± 1.08	4.52 ± 1.47	41.88 ± 5.25	6.52 ± 1.34
	2019	17.37 ± 2.27	4.74 ± 0.95	5.76 ± 1.33	1.86 ± 0.53	29.37 ± 5.16	5.41 ± 0.41
**PN**	2018	1.45 ± 0.15	11.47 ± 0.81	1.56 ± 0.09	6.96 ± 0.25	40.58 ± 2.12	1.98 ± 0.14
	2019	**6.10 *** ± 1.37	13.85 ± 3.10	3.04 ± 0.85	**4.87 *** ± 0.74	48.44 ± 10.26	2.51 ± 0.58
**SO**	2018	28.22 ± 3.39	0.89 ± 0.05	5.48 ± 2.49	1.46 ± 0.07	7.64 ± 0.12	5.76 ± 0.88
	2019	73.37 * ± 2.48	2.41 ± 0.68	**16.58 *** ± 1.50	3.70 ± 1.43	21.87 ± 9.41	16.82 ± 6.05
**TE**	2018	41.04 ± 11.67	8.21 ± 4.00	6.59 ± 3.01	5.57 ± 1.92	34.01 ± 16.29	6.70 ± 1.49
	2019	35.70 ± 0.15	**19.47 *** ± 2.29	4.01 ± 0.65	9.03 ± 0.71	**65.25 *** ± 5.43	6.20 ± 0.20
**XA**	2018	5.47 ± 0.36	3.20 ± 1.01	3.73 ± 0.81	5.53 ± 0.40	39.05 ± 2.68	2.46 ± 0.05
	2019	7.88 ± 1.13	0.72 * ± 0.02	1.34 ± 1.58	**15.23 *** ± 1.74	116.27 * ± 10.88	1.09 ± 1.88
**ZA**	2018	6.31 ± 0.90	17.62 ± 3.57	3.15 ± 0.58	10.95 ± 1.83	71.60 ± 15.48	4.67 ± 1.37
	2019	8.75 ± 3.28	**0.32 *** ± 0.32	2.03 ± 0.74	10.53 ± 1.57	85.14 ± 8.86	4.21 ± 0.73
**Variety**		***	***	***	***	***	***
**% Variation**		79.33	49.67	77.14	63.40	64.01	73.32
**Year**		**	ns	ns	**	***	ns
**% Variation**		8.25	0.16	0.00	7.67	15.97	2.51
**Variety x Year**		***	***	***	***	***	***
**% Variation**		9.55	39.65	16.33	17.41	12.73	14.05

My-Glc: myricetin3-O-glucoside; Iso-Rut: isorhamnetin 3-rutinoside; Kaemp-Glc: kaempherol 3-glucoside; Quer-Rut: quercetin 3-O-rutinoside-7-O-glucoside; Quer-Glc: quercetin-3-O-glucoside; Quer-GlcU: quercetin 3-glucuronide. For ANOVA and factorial analysis: ns = non-significant; ** *p* < 0.01; *** *p* < 0.001. Within each variety, * indicate significant differences among years after Student’s *t*-test. Data are reported as mean (*n* = 6). See Table 1 for varieties abbreviations.

**Table 4 plants-12-00004-t004:** Concentration of non-anthocyanin phenolic compounds: flavanols (FLAVA) from the EVEGA germplasm bank in 2018 and 2019 vintages. (Values are expressed as mg kg^−1^ FW).

Variety	Year	Catechin	Epicatechin	Gallocatechin	Epicatechin Gallate	Σ Cat	PB1	PB2	Σ PAC
**AT**	2018	42.75 ± 2.16	14.77 ± 0.49	27.03 ± 4.73	3.08 ± 0.33	87.63 ± 3.39	39.18 ± 2.92	17.74 ± 4.30	56.92 ± 1.38
	2019	50.04 ± 15.55	18.88 ± 7.23	76.36 * ± 17.43	38.48 ± 20.95	**183.77 *** ± 32.98	53.12 ± 8.85	13.78 ± 1.90	66.90 ± 9.17
**BR**	2018	65.21 ± 7.54	47.44 ± 5.33	25.92 ± 1.66	13.22 ± 0.55	151.79 ± 15.09	44.27 ± 9.67	23.62 ± 3.53	67.90 ± 13.20
	2019	94.64 ± 14.66	71.63 ± 11.07	**112.30 *** ± 18.29	**37.09 *** ± 2.85	**315.66 *** ± 46.24	62.02 ± 19.94	32.67 ± 6.79	94.70 ± 26.71
**CB**	2018	67.52 ± 0.70	37.70 ± 0.81	51.46 ± 4.50	6.52 ± 0.54	163.19 ± 6.56	58.35 ± 11.91	47.01 ± 8.05	105.36 ± 3.87
	2019	81.48 * ± 3.51	**64.38 *** ± 5.64	102.97 * ± 6.46	10.17 * ± 1.02	**259.00 *** ± 15.00	55.81 ± 2.37	53.07 ± 3.78	108.89 ± 6.05
**CL1**	2018	72.36 ± 19.02	43.29 ± 17.32	29.32 ± 8.44	8.44 ± 3.88	153.40 ± 48.67	47.34 ± 7.25	23.87 ± 0.01	71.22 ± 7.27
	2019	96.93 ± 11.11	88.24 ± 16.04	**125.16 *** ± 14.33	**25.85 *** ± 3.63	**336.18 *** ± 42.87	63.14 ± 6.35	32.42 * ± 1.65	**95.56 *** ± 7.99
**CL2**	2018	67.61 ± 3.11	33.13 ± 6.21	37.67 ± 1.85	28.81 ± 28.48	167.22 ± 17.30	50.86 ± 8.15	32.46 ± 9.47	83.32 ± 1.31
	2019	**111.15 *** ± 3.56	**54.47 *** ± 2.61	143.42 * ± 8.05	20.56 ± 2.01	**329.61 *** ± 11.10	**78.64 *** ± 7.85	43.64 ± 1.36	**122.27 *** ± 8.31
**CT**	2018	178.48 ± 31.50	141.31 ± 20.15	63.83 ± 7.74	17.61 ± 1.71	401.23 ± 61.11	75.01 ± 12.23	52.61 ± 9.99	127.62 ± 22.21
	2019	95.86* ± 22.45	69.28 * ± 7.84	118.15 ± 25.66	11.29 ± 2.94	294.58 ± 57.81	57.43 ± 1.93	42.85 ± 19.70	100.27 ± 19.08
**CS**	2018	44.22 ± 4.72	12.14 ± 2.77	27.47 ± 5.06	3.50 ± 0.07	87.33 ± 12.62	38.75 ± 6.11	13.15 ± 4.56	51.90 ± 10.67
	2019	**129.99 *** ± 29.49	55.32 ± 18.28	**155.18 *** ± 32.72	**21.32 *** ± 0.33	**361.80 *** ± 80.39	90.23 * ± 14.31	**31.40 *** ± 1.02	**121.63 *** ± 13.88
**CO**	2018	328.51 ± 24.00	293.41 ± 14.97	57.84 ± 1.47	21.79 ± 0.68	701.56 ± 38.17	92.13 ± 3.87	70.29 ± 4.13	162.42 ± 0.27
	2019	425.73 * ± 18.40	277.59 ± 10.65	471.36 * ± 17.66	19.66 ± 5.55	**1194.34 *** ± 44.47	**126.83 *** ± 7.51	68.48 ± 2.56	**195.31 *** ± 9.99
**ES**	2018	21.24 ± 0.68	7.46 ± 0.03	22.62 ± 3.14	11.15 ± 2.06	62.48 ± 5.85	10.19 ± 0.89	67.05 ± 13.70	77.24 ± 14.59
	2019	**38.98 *** ± 6.27	**14.53 *** ± 2.82	**62.48 *** ± 6.84	**23.50 *** ± 1.27	**139.48 *** ± 14.47	**37.78 *** ± 2.96	**16.98 *** ± 2.54	54.76 ± 3.10
**EV3**	2018	431.88 ± 56.17	160.01 ± 8.94	72.71 ± 3.73	33.22 ± 5.27	697.82 ± 74.11	238.95 ± 48.57	91.71 ± 5.81	330.67 ± 54.38
	2019	384.27 ± 40.42	126.48 ± 13.12	420.39 * ± 39.17	19.23 * ± 3.64	**950.37 *** ± 88.51	97.28 * ± 10.19	42.83 * ± 2.86	**140.11 *** ± 11.73
**EV4**	2018	92.86 ± 12.36	49.21 ± 11.42	31.39 ± 6.69	5.58 ± 2.57	179.04 ± 33.03	70.12 ± 5.81	38.18 ± 0.77	108.30 ± 6.58
	2019	110.79 ± 35.68	45.57 ± 16.30	**143.19 *** ± 36.14	**28.25 *** ± 2.68	327.80 ± 86.72	92.47 ± 10.64	33.84 ± 6.51	**126.31 *** ± 4.28
**EV6**	2018	373.78 ± 33.65	120.54 ± 8.53	68.98 ± 1.00	10.17 ± 10.41	573.47 ± 30.77	211.04 ± 23.26	64.17 ± 8.44	275.22 ± 31.70
	2019	359.63 ± 14.92	139.02 ± 6.97	399.49 * ± 11.92	14.99 ± 1.23	**913.13 *** ± 33.52	**104.14 *** ± 27.17	**37.68 *** ± 6.70	**141.82 *** ± 33.85
**FE**	2018	53.21 ± 0.41	36.31 ± 6.00	25.96 ± 3.36	12.14 ± 1.98	127.62 ± 7.80	54.40 ± 0.83	35.58 ± 2.12	89.98 ± 1.29
	2019	**36.86 *** ± 5.63	**20.80 *** ± 4.10	**59.24 *** ± 11.85	**38.75 *** ± 4.58	155.64 ± 25.86	62.91 ± 11.53	25.09 ± 4.69	88.00 ± 16.21
**GA**	2018	45.65 ± 3.58	21.71 ± 1.91	22.13 ± 1.26	2.29 ± 0.64	91.78 ± 7.40	27.54 ± 5.69	16.22 ± 0.75	43.76 ± 6.45
	2019	42.45 ± 1.53	23.27 ± 1.46	**56.87 *** ± 8.13	**26.54 *** ± 6.06	**149.13 *** ± 12.78	34.03 ± 6.79	**12.69 *** ± 1.12	46.72 ± 7.86
**GN**	2018	25.17 ± 0.04	19.17 ± 1.53	16.23 ± 0.38	17.06 ± 7.45	77.63 ± 6.34	27.55 ± 0.81	14.20 ± 0.25	41.75 ± 0.57
	2019	20.07 * ± 1.55	15.66 * ± 0.95	27.77 ± 9.53	24.58 ± 2.97	88.08 ± 12.96	29.65 ± 1.36	12.39 ± 1.23	42.05 ± 2.33
**HI**	2018	46.69 ± 12.20	12.39 ± 4.51	16.85 ± 9.42	1.46 ± 0.54	77.39 ± 26.67	22.40 ± 1.66	7.79 ± 3.70	30.19 ± 5.36
	2019	62.04 ± 14.35	19.04 ± 3.08	**85.98 *** ± 20.30	**40.94 *** ± 2.22	**208.01 *** ± 37.08	**44.49 *** ± 3.12	10.69 ± 1.08	**55.18 *** ± 3.12
**MA**	2018	41.96 ± 16.22	12.07 ± 0.95	21.99 ± 2.92	11.10 ± 5.57	87.12 ± 23.76	28.38 ± 6.55	12.74 ± 0.20	41.12 ± 6.75
	2019	33.95 ± 10.78	13.34 ± 4.02	48.25 ± 11.95	23.72 ± 4.00	119.26 ± 25.84	32.86 ± 3.44	11.80 ± 2.41	44.66 ± 5.80
**ME**	2018	35.62 ± 4.13	14.33 ± 4.25	19.01 ± 0.00	8.83 ± 0.20	77.79 ± 8.58	32.89 ± 0.75	12.04 ± 1.23	44.93 ± 0.48
	2019	42.36 ± 1.33	21.18 ± 2.10	**67.66 *** ± 3.71	**17.63 *** ± 1.02	**148.82 *** ± 5.69	**50.30 *** ± 2.76	11.80 ± 1.06	**62.10 *** ± 3.71
**MZ**	2018	62.93 ± 15.91	14.50 ± 2.21	14.75 ± 7.33	2.76 ± 0.72	94.94 ± 26.18	41.15 ± 20.89	12.48 ± 7.61	53.63 ± 28.50
	2019	72.89 ± 38.16	19.36 ± 11.72	92.80 ± 48.11	22.73 * ± 3.19	207.78 ± 100.74	53.67 ± 17.48	13.84 ± 4.99	67.51 ± 21.92
**MH**	2018	92.98 ± 35.92	45.85 ± 25.94	14.59 ± 12.82	2.31 ± 1.38	155.73 ± 76.05	41.35 ± 7.59	21.29 ± 14.32	62.64 ± 21.91
	2019	82.58 ± 9.87	37.82 ± 7.53	**116.87 *** ± 15.29	**16.98 *** ± 5.28	254.25 ± 33.31	**65.88 *** ± 6.13	18.34 ± 1.92	84.22 ± 7.82
**MO**	2018	26.45 ± 0.70	20.10 ± 1.42	10.94 ± 4.71	2.40 ± 0.43	59.89 ± 7.26	21.95 ± 0.00	15.05 ± 3.08	37.00 ± 3.08
	2019	29.89 ± 2.92	21.88 ± 1.80	49.57 * ± 3.23	**31.98 *** ± 3.41	**133.32 *** ± 5.15	**36.24 *** ± 3.17	15.92 ± 2.51	**52.16 *** ± 4.86
**PC**	2018	80.86 ± 3.10	27.14 ± 1.78	37.87 ± 1.92	28.65 ± 15.76	174.51 ± 12.80	43.73 ± 21.33	15.38 ± 3.12	59.12 ± 24.44
	2019	**135.55 *** ± 9.72	**77.40 *** ± 3.26	156.39 ± 9.34	13.95 ± 8.41	**383.28 *** ± 24.18	40.73 ± 1.61	20.19 ± 1.94	60.92 ± 2.81
**PE**	2018	106.97 ± 8.42	75.08 ± 18.60	33.87 ± 10.96	9.18 ± 5.33	225.10 ± 32.66	76.28 ± 8.73	49.33 ± 4.33	125.61 ± 13.06
	2019	118.59 ± 8.35	**128.93 *** ± 12.84	**142.79 *** ± 10.25	11.64 ± 2.09	**401.95 *** ± 23.22	99.13 ± 12.59	59.77 ± 6.87	158.90 ± 19.43
**PN**	2018	54.59 ± 5.20	25.74 ± 2.67	24.76 ± 7.80	1.66 ± 0.17	106.75 ± 15.84	34.04 ± 11.61	22.05 ± 2.21	56.09 ± 13.82
	2019	70.00 ± 11.14	50.00 * ± 9.94	86.39 * ± 16.95	**28.30 *** ± 5.70	**234.69 *** ± 42.16	45.47 ± 8.20	25.16 ± 4.59	70.63 ± 12.50
**SO**	2018	29.96 ± 2.23	14.91 ± 3.73	25.98 ± 3.34	2.47 ± 0.63	73.33 ± 9.93	37.61 ± 6.52	25.34 ± 3.95	62.95 ± 10.46
	2019	38.08 ± 6.31	22.78 ± 2.09	**70.17 *** ± 8.70	9.90 ± 4.89	**140.93 *** ± 19.39	**55.97 *** ± 2.80	26.35 ± 1.58	82.32 ± 3.89
**TE**	2018	30.95 ± 5.46	10.25 ± 0.17	13.84 ± 3.64	14.09 ± 11.50	69.13 ± 20.76	28.10 ± 8.54	14.74 ± 1.92	42.83 ± 10.46
	2019	33.30 ± 2.86	**15.51 *** ± 1.62	53.14 * ± 3.26	27.46 ± 1.71	**129.41 *** ± 6.47	37.46 ± 2.57	13.66 ± 0.89	51.12 ± 2.71
**XA**	2018	259.62 ± 2.50	112.37 ± 9.65	67.34 ± 7.64	4.09 ± 0.34	443.42 ± 4.85	85.06 ± 7.28	46.56 ± 4.97	131.61 ± 2.31
	2019	**331.34 *** ± 13.77	117.72 ± 1.50	**368.50 *** ± 10.71	46.33 * ± 6.05	**863.88 *** ± 31.05	**112.27 *** ± 2.92	43.83 ± 3.05	**156.10 *** ± 5.74
**ZA**	2018	140.58 ± 57.24	91.30 ± 24.08	34.06 ± 6.83	2.72 ± 0.88	268.66 ± 89.03	52.24 ± 9.70	34.10 ± 6.40	86.34 ± 16.11
	2019	**328.22 *** ± 17.58	**196.06 *** ± 15.52	**366.72 *** ± 4.05	**36.53 *** ± 4.22	**927.52 *** ± 28.74	**93.76 *** ± 12.94	**49.86 *** ± 2.75	**143.62 *** ± 15.17
**Variety**		***	***	***	ns	***	***	***	***
**% Variation**		93.67	91.99	50.49	20.91	82.50	70.34	78.20	74.93
**Year**		ns	ns	***	***	***	ns	ns	ns
**% Variation**		0.66	0.77	25.17	30.75	8.36	0.67	0.50	0.12
**Variety x Year**		***	***	***	***	***	***	***	***
**% Variation**		4.15	5.86	23.16	34.16	7.80	24.03	16.29	20.76

Σ Cat: total catechins including catechin, epicatechin, gallocatechin, and epicatechin gallate; PB1: procyanidin B1; PB2: procyanidin B2; Σ PAC: total procyanidins including PB1 and PB2. For ANOVA and factorial analysis: ns = non-significant; *** *p* < 0.001. Within each variety, * indicate significant differences among years after Student’s *t*-test. Data are reported as mean (*n* = 6). See Table 1 for varieties abbreviations.

**Table 5 plants-12-00004-t005:** Concentration of hydroxybenzoic and hydroxycinnamic acids from the EVEGA germplasm bank in 2018 and 2019 vintages. (Values are expressed as mg·kg^−1^ FW).

Variety	Year	GA	CF	FR	COU
**AT**	2018	0.77 ± 0.02	0.25 ± 0.08	0.00 ± 0.00	0.91 ± 0.21
	2019	0.56 ± 0.15	0.50 ± 0.16	0.69 ± 0.61	1.11 ± 0.11
**BR**	2018	0.94 ± 0.18	1.14 ± 1.13	0.00 ± 0.00	2.05 ± 0.49
	2019	0.95 ± 0.77	1.91 ± 0.75	0.01 ± 0.02	**0.00 *** ± 0.00
**CB**	2018	2.19 ± 0.59	0.82 ± 0.89	0.00 ± 0.00	1.53 ± 0.34
	2019	**1.15 *** ± 0.05	1.54 ± 0.04	**0.03 *** ± 0.01	0.54 ± 0.47
**CL1**	2018	1.76 ± 0.15	0.14 ± 0.08	0.10 ± 0.08	1.28 ± 0.53
	2019	1.34 ± 0.31	**0.80 *** ± 0.18	0.10 ± 0.03	1.09 ± 0.13
**CL2**	2018	1.66 ± 0.15	0.38 ± 0.27	0.09 ± 0.13	1.48 ± 0.04
	2019	**0.60 *** ± 0.05	**2.57 *** ± 0.24	0.08 ± 0.08	1.41 ± 0.20
**CT**	2018	3.28 ± 0.51	0.09 ± 0.05	0.00 ± 0.00	1.62 ± 0.71
	2019	**1.01 *** ± 0.18	**1.49 *** ± 0.14	0.07 ± 0.07	0.54 ± 0.47
**CS**	2018	1.49 ± 0.07	1.42 ± 1.18	0.54 ± 0.65	1.16 ± 0.25
	2019	1.87 ± 0.67	1.28 ± 0.10	0.13 ± 0.13	1.36 ± 0.15
**CO**	2018	5.88 ± 0.24	2.06 ± 0.56	0.50 ± 0.13	1.85 ± 0.27
	2019	6.15 ± 1.24	**0.71 *** ± 0.14	0.39 ± 0.03	**1.24 *** ± 0.09
**ES**	2018	1.33 ± 1.26	0.54 ± 0.17	0.00 ± 0.00	0.96 ± 0.08
	2019	1.46 ± 0.40	**2.23 *** ± 0.38	1.08 ± 0.48	0.76 ± 0.28
**EV3**	2018	10.98 ± 1.60	0.24 ± 0.13	0.92 ± 0.22	0.91 ± 0.16
	2019	**3.65 *** ± 0.19	**1.14 *** ± 0.18	**0.06 *** ± 0.03	1.38 ± 0.19
**EV4**	2018	3.08 ± 0.40	0.23 ± 0.07	0.43 ± 0.07	1.49 ± 0.37
	2019	**1.27 *** ± 0.27	0.28 ± 0.19	**0.01 *** ± 0.01	0.86 ± 0.75
**EV6**	2018	6.44 ± 0.38	0.10 ± 0.02	0.98 ± 0.24	2.27 ± 1.44
	2019	**3.24 *** ± 0.22	**0.67 *** ± 0.07	**0.32 *** ± 0.02	0.66 ± 0.58
**FE**	2018	1.90 ± 0.43	0.30 ± 0.08	0.00 ± 0.00	2.24 ± 1.89
	2019	**0.90 *** ± 0.20	0.45 ± 0.08	0.01 ± 0.01	0.55 ± 0.48
**GA**	2018	0.41 ± 0.07	0.65 ± 0.21	0.00 ± 0.00	0.96 ± 0.14
	2019	1.29 ± 1.63	1.10 ± 0.18	0.05 ± 0.05	0.75 ± 0.18
**GN**	2018	1.47 ± 0.47	0.20 ± 0.02	0.08 ± 0.11	1.02 ± 0.00
	2019	**0.53 *** ± 0.03	0.30 ± 0.08	0.04 ± 0.01	**0.50 *** ± 0.04
**HI**	2018	1.00 ± 0.14	0.10 ± 0.14	0.00 ± 0.00	1.13 ± 0.07
	2019	1.12 ± 0.16	0.89 ± 0.39	0.03 ± 0.06	1.31 ± 0.54
**MA**	2018	1.28 ± 0.45	0.45 ± 0.04	0.28 ± 0.40	1.15 ± 0.56
	2019	0.83 ± 0.13	**1.17 *** ± 0.18	0.00 ± 0.00	1.19 ± 0.18
**ME**	2018	0.42 ± 0.19	0.19 ± 0.05	0.34 ± 0.48	0.99 ± 0.06
	2019	0.21 ± 0.03	0.78 ± 0.60	0.00 ± 0.00	1.11 ± 0.11
**MZ**	2018	4.08 ± 2.18	0.03 ± 0.00	0.22 ± 0.31	1.24 ± 0.31
	2019	2.14 ± 0.82	**0.31 *** ± 0.01	0.01 ± 0.01	0.50 ± 0.87
**MH**	2018	5.41 ± 0.97	0.00 ± 0.00	0.05 ± 0.07	1.17 ± 0.53
	2019	**1.63 *** ± 1.20	0.55 * ± 0.17	0.15 ± 0.13	1.25 ± 0.10
**MO**	2018	1.56 ± 0.93	0.39 ± 0.32	0.00 ± 0.00	1.15 ± 0.30
	2019	0.56 ± 0.11	0.33 ± 0.12	0.40 ± 0.69	1.52 ± 0.16
**PC**	2018	1.25 ± 0.18	0.61 ± 0.54	0.00 ± 0.00	1.10 ± 0.07
	2019	1.01 ± 0.33	0.16 ± 0.06	0.50 ± 0.52	**0.91 *** ± 0.03
**PE**	2018	3.51 ± 0.86	0.17 ± 0.18	0.92 ± 0.02	0.97 ± 0.31
	2019	**1.18 *** ± 0.26	**1.34 *** ± 0.10	**0.29 *** ± 0.08	**1.66 *** ± 0.15
**PN**	2018	2.70 ± 1.02	0.00 ± 0.00	0.00 ± 0.00	1.04 ± 0.24
	2019	1.52 ± 0.13	**0.44 *** ± 0.10	0.04 ± 0.04	0.52 ± 0.45
**SO**	2018	2.24 ± 1.35	0.08 ± 0.06	0.00 ± 0.00	0.91 ± 0.20
	2019	0.99 ± 0.00	**1.65 *** ± 0.06	2.22 ± 1.93	0.90 ± 0.78
**TE**	2018	1.60 ± 0.25	0.24 ± 0.04	0.21 ± 0.30	0.88 ± 0.19
	2019	**0.76 *** ± 0.12	**0.86 *** ± 0.23	0.09 ± 0.02	0.68 ± 0.59
**XA**	2018	3.63 ± 0.22	1.49 ± 1.60	0.14 ± 0.04	1.13 ± 0.07
	2019	**2.01 *** ± 0.51	1.04 ± 0.14	0.18 ± 0.05	0.83 ± 0.76
**ZA**	2018	3.22 ± 1.90	1.25 ± 0.26	0.24 ± 0.32	0.85 ± 0.01
	2019	1.97 ± 0.70	0.68 ± 0.25	0.29 ± 0.09	0.24 ± 0.42
**Variety**		***	***	**	ns
**% Variation**		64.53	43.27	32.58	19.14
**Year**		***	***	ns	***
**% Variation**		10.08	12.44	0.20	9.64
**Variety x Year**		***	***	***	**
**% Variation**		17.54	27.83	32.81	31.18

GA: gallic acid; CF: caffeic acid; FR: ferulic acid; COU: coutaric acid. For ANOVA and factorial analysis: ns = non-significant; ** *p* < 0.01; *** *p* < 0.001. Within each variety, * indicate significant differences among years after Student’s *t*-test. Data are reported as mean (*n* = 6). See Table 1 for varieties abbreviations.

## Data Availability

Data is contained within the article.

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
