# Peer review of "Non-Anthocyanin Compounds in Minority Red Grapevine Varieties Traditionally Cultivated in Galicia (Northwest Iberian Peninsula), Analysis of Flavanols, Flavonols, and Phenolic Acids"

_plants, 2022, doi:10.3390/plants12010004_

Round 1

Reviewer 1 Report

The manuscript entitled “Non anthocyanin compounds in minority red grapevine varieties traditionally cultivated in Galicia (Northwest Iberian Peninsula). Analysis of flavanols, flavonols and phenolic acids” established the non-anthocyanin profiles of 28 grapevine varieties and evaluated the potentiality of these compounds be used as varietal markers.

The article contains very interesting data, however, the way in which the authors present and discuss the results is very extensive and described. In my opinion, the results section should be rewritten to compare the results and highlight the significant differences between the two vintages, and among the studied varieties. Instead of just commenting on who has less and more content of the compounds under study.

All abbreviation should be described when reported for the first time.

Figures resolution must be improved.

Abstract: 16 NAN should be sixteen NAN

According to the authors “Absorbances at 280, 320, 360, and 520 nm were measured by the DAD detector and….” Why the authors considered 520 nm?

Quercetine-3-glucoside should be quercetin-3-glucoside

Please provide the data related to method validation.

Author Response

We appreciate the valuable comments of the Reviewer and we have made changes following reviewer’s suggestions. We hope the paper would be now suitable for publication in Plants’ journal.

  1. The article contains very interesting data, however, the way in which the authors present and discuss the results is very extensive and described. In my opinion, the results section should be rewritten to compare the results and highlight the significant differences between the two vintages, and among the studied varieties. Instead of just commenting on who has less and more content of the compounds under study

Response: Thank you for your review in detail.

  1. Respect to comparison between years we should indicate:
  • The variation due to year of each compound in each variety has been calculated and the variation percentages attributed to ‘year’ and ‘variety’. The year factor has been calculated for each compound and variety as it appears in Tables 3, 4 and 5. In the revised version of the article, the statistically different values for each compound and variety have been indicated in bold for greater visibilit In addition, the PCA analysis has been carried out with the results of both years to highlight the differences due to this factor.

References to significant differences between the two vintages were written in the original manuscript. They appear now in the following pages and paragraphs of the revised manuscript:

  • Page 15, second paragraph, lines 296-306:

On the other hand, as a general trend, every FLAVO compound, except for Kaemp-Glc, showed higher contents in 2019 vintage. This effect had a significative effect in My-Glc, Quer-Glc and Quer-Rut. The highest interannual variations were registered in My-Glc (64.70 %) and Quer-Glc (61.92 %). Besides, results from Table 3 reflect that, when FLAVO compounds were analyzed on a year-by-year basis, the number of significant in-terannual differences depended on the variety considered: ‘Albarín Tinto’, ‘Caiño Bravo’, ‘Caiño Longo 2’, ‘Castañal’, ‘Corbillón’, ‘Espadeiro’, ‘Ferrón’, ‘Garnacha’, ‘Gran Negro’, ‘Mandón’, ‘Mencía’, ‘Merenzao’, ‘Moscatel de Hamburgo’, ‘Mouratón’, ‘Pan y Carne’, ‘Pedral’, ‘Picapoll Negro’, ‘Sousón’, ‘Tempranillo’ and ‘Zamarrica’ were those varieties that showed interannual significant differences for a lower number of compounds, looking as the more stable varieties in terms of FLAVO compounds.

  • Page 22, third paragraph, lines 406-414

It details that it was seen a ‘year’ effect in different flavanol compounds as well as there were mentioned those varieties that showed significant differences among years for a lower number of compounds and as a result, which could be initially more stable among years.

As it can be seen, except for PB2, every FLAVA compound showed higher values in 2019. ANOVA found a significant effect of the ‘year’ in Galo, GaloEpi, and sum of catechins (Σ Cat). ‘Year’ effect explained a low percentage of the variance (around 25-30 %, in both compounds). On the other hand, it should be noted that different varieties were affected by the ‘year’ effect in a different way. Thus ‘Albarín Tinto’, ‘Brancellao’, ‘Caiño Tinto’, ‘Evega 4’, ‘Evega 6’, ‘Gran Negro’, ‘Mandón’, ‘Merenzao’, ‘Pan y Carne’, ‘Pedral’, ‘Sousón’ and ‘Tempranillo’ were those varieties that showed interannual significant differences among years for a lower number of FLAVA compounds, what could initially be interpreted as more regular varieties among years.

  1. Respect to comparison between varieties we should indicate:

The figures 2, 3 and 4 in the original and review manuscript reflect flavonol, flavanol and phenolic acids profiles and clearly allows to compare within varieties among years as well as compare among different varieties trying to avoid the ‘year’ effect. Besides, a detailed study of those varieties that, in agreement with flavanol, flavanol and phenolic acids profiles was written.

On other hand, since the objective of the article is to characterize the minority varieties and to facilitate this characterization, sentences about this concept were given in the original manuscript.

They appear now in the following pages, paragraphs and lines of the revised manuscript:

  • Page 11, second paragraph, lines 257-259

In terms of decreasing year variability, and to characterize varieties, the percentage of NAN and AN respect to the TPC were calculated.

  • Page 22, first paragraph, lines 382-397

Since a wide range of concentrations can be observed in every compound, the ‘variety’ factor was significant and explained a big percentage of variation (> 50 %) in all of them, except for GaloEpi. When the sum of monomers Σ (Cat, Epi, Galo, and GaloEpi), is considered in 2018, ‘Mouratón’, considered as an ANV, registered the lowest value (59.89) while ‘Corbillón’, classified as an NANV, registered the highest one (701.56 mg·kg-1 FW). In 2019, the values ranged from 88.08 in ‘Gran Negro’ up to 1194 in ‘Corbillón’. The minimum procyanidin total values, as Σ (PB1, PB2), corresponded to ‘Híbrido’ (30.19) and the maximum to ‘Evega 3’ (330.67) while in 2019 corresponded to ‘Gran Negro’ (42.05) and ‘Corbillón’ (195.31). Liang et al. [47] found mean values of 31, 10 and 4 mg·kg-1 FW for Cat, Epi, GaloEpi and 9.4 and 2 for PB1 and PB2 when polyphenolic profiles in the berry samples of 344 European grape (Vitis vinifera L.) cultivars were evaluated after removing all seeds for two consecutive years. The grapes of cv ‘Tempranillo’, the reference variety in this work, registered values of 69.13 and 129.41 for the sum of monomers and 42.83 and 51.12 as dimers in 2018 and 2019 respectively. Thus ‘Corbillón’ and ‘Evega 3’ were characterized by the highest FLAVA values of monomers and dimers. These facts must be considered in the winemaking process of these varieties.

  • Page 25, first paragraph, lines 441-449

Here, a detailed study about the varieties with higher gallic acid contents, as the major phenolic acid detected in almost every variety, was written. Besides, varieties with the highest and lowest caffeic and coutaric acid values were also detailed, and finally varieties with higher hydroxycinnamic acid compounds were highlighted.

Among the acids analyzed, the hydroxybenzoic acid gallic (GA), was the major compound in all varieties in both years, except for ‘Albarín Tinto’ and ‘Merenzao’, where coutaric acid (COU) reached the highest values. In 2018 and 2019 years, the highest values of GA were registered in ‘Corbillón’ (5.88 and 6.15 mg·kg-1 FW), ‘Evega3’ (10.98 and 3.65) and ‘Evega 6’ (6.44 and 3.24). According to Garrido and Borges [29], GA is usually the most abundant substance of this group, and it is described as the most important phenolic compound since it is the precursor of all hydrolyzable tannins and is encompassed in condensed tannins. After it, COU was the most abundant substance belonging to phenolic acids identified and quantified, with a minimum in ‘Zamarrica’ (0.85 mg·kg-1 FW) and ‘Tempranillo’ (0.88) in 2018, and in ‘Brancellao’ (0.0) and ‘Zamarrica’ (0.24) in 2019. In front, ‘Evega 6’ (2.27) and ‘Pedral’ (1.66) registered the maximum values in 2018 and 2019 years. Finally, respect to CF, it is noticed that there were generally higher values in 2019, being ‘Caiño Longo 2’ and ‘Espadeiro’ those varieties with maximum values. Respect to FR, it was the acid that showed the smallest quantities, with ‘Sousón’ and ‘Espadeiro’ recording the highest values in 2019. Thus, regarding to the global hydroxycinnamic compounds the richest varieties were ‘Corbillón’ in 2018 (4.41) and ‘Sousón’ in 2019 (4.78).

  • Page 26, first paragraph, lines 472-478

The varieties with lower interannual differences were mentioned.

Furthermore, ‘Albarín Tinto’, ‘Brancellao’, ‘Caiño Longo 1’, ‘Castañal’, ‘Espadeiro’, ‘Ferrón’, ‘Garnacha’, ‘Híbrido’, ‘Mandón’, ‘Mencía’, ‘Merenzao’, ‘Mouratón’, ‘Pan y Carne’, ‘Picapoll Negro’, ‘Sousón’, ‘Xafardán’ and ‘Zamarrica’ were those varieties that showed interannual significant differences for a fewer number of acid compounds, what a priori could be associated with more regularity among years. There were also found significant differences for the interaction ‘variety x year’ on the values of every compound.

  1. All abbreviation should be described when reported for the first time.

Response: Thank you for your review in detail. In order to prevent the abbreviations NANV, ANV and NANAV shown in the summary from generating confusion, the sentence has been rewritten as follows (In Lines 21-24 in blue in the revised manuscript):

The percentage of total NAN respect to the total polyphenol content (TPC) values was calculated for each sample being established three categories: those high percentage NAN varieties (NANV), those varieties showing low percentage of NAN (ANV) and finally those varieties showing medium percentages of NAN (NANAV).

  1. Figures resolution must be improved

Response: Thank you for your advice. We have improved figures´ quality modifying their resolution or changing colours to make the data look clearer, we hope they are now much easier to read.

  1. Abstract: 16 NAN should be sixteen NAN

Response: Thank you for your advice. The correction is carried out and sixteen was written in the new manuscript (Line 18).

  1. According to the authors “Absorbances at 280, 320, 360, and 520 nm were measured by the DAD detector and….” Why the authors considered 520 nm?

Response: Thank you for your detailed revision.

Since total value of anthocyanins is indicated in the work and these compounds are identified and quantified at 520 nm, this absorbance was included in the methodology of the manuscript In any case, sentence was rewrite in order a major comprehension. (Lines 184-186 of the revised manuscript):

Absorbances at 320, 360, and 520 nm were measured by the DAD detector to quantify phenolics acids, flavonols and anthocyanin and excitation at 280 and emission at 320 nm were measured by the FLD detector to flavanols compounds.

  1. Quercetine-3-glucoside should be quercetin-3-glucoside

Response: Thank you for your advice. It has already been corrected.

Quercetin-3- glucoside was written in the revised manuscript (line 197)

  1. Please provide the data related to method validation

Response: Since the method has been published in Scientific Journal, some performance characteristics of the method were established in our laboratory as linearity. The results obtained for the linearity assessment are shown in the Table in “Calibration HPLC” file attached.

Reviewer 2 Report

The aim of the study was to evaluate the content of non-anthocyanin compounds in 28 grape varieties. The research is not very innovative, but taking into account consumer expectations regarding the quality of wine, these experiences have an important practical character. The manuscript is carefully written, the results described and discussed with the latest literature. The obtained results are analyzed by an appropriate statistical method. A few remarks concern the innovativeness of the work, the readability of charts and analytical details. They are presented below:

The purpose of the work is clearly presented and well justified. However, the introduction lacks confirmation whether the research conducted on these 28 grapevine varieties is innovative. Were the content of polyphenolic compounds in the tested cultivars assessed in previous publications?

The description of the sample preparation and the analytical methods used require more detailed information. The grape samples were frozen at -20 degrees and then extracted with the extraction mixture. Were the fruits thawed, or was homogenization performed on frozen fruits? If the sample was thawed, how? 50 g of fruit were taken for extraction. How were the fruits divided to measure 50 g? Cut the fruit into quarters? So that the ratio of peel, pulp and seeds is the same as in the fruit? Was about 50 g of fruit weighed? This should be specified.

I also recommend writing the gradient in the HPLC method, even if it is given in the cited publication.

The paper states that the identification of polyphenolic compounds was made on the basis of their elution order and by comparison with the retention times of commercially available standards. Of course, a more adequate tool to confirm the correct identification would be the use of LC-MS/MS, but why wasn't at least a comparison of the spectra with the available standards?

I recommend to improve the readability of figure number 4. The legend is especially hard to read. A similar remark applies to charts 6-8. They are very unreadable.

Because one of the reasons for the research was to assess whether the wine obtained from a given variety will undergo co-pigmentation during storage, which will improve the color stability. Therefore, I believe that the conclusions should indicate a further direction of research related to the fermentation process of fruits with the highest content of polyphenolic compounds and check whether the actual color of such wines will be more stable.

Author Response

REVIEWER 2.

We appreciate the valuable comments of the Reviewer. We have made changes following reviewer’s suggestions. We hope the paper would be now suitable for publication in Plants´ journal.

  1. The purpose of the work is clearly presented and well justified. However, the introduction lacks confirmation whether the research conducted on these 28 grapevine varieties is innovative. Were the content of polyphenolic compounds in the tested cultivars assessed in previous publications?

Response: The anthocyanin characterization of these varieties has been provided by a previous research: Díaz-Fernández, Á., Díaz-Losada, E., Moreno, D., & Esperanza Valdés Sánchez, M. (2022). Anthocyanin profile of Galician endangered varieties. A tool for varietal selection. Food Research International, 154. https://doi.org/10.1016/j.foodres.2022.110983. However, no individual study was made of non-anthocyanin compounds. To the best of our knowledge this is the first time that this detailed description has been made for most of the varieties studied in this manuscript and this work contribute to a better knowledge of them. This concept is mentioned in the section conclusions (Lines 620-624 in the revised manuscript):

In this study, flavanol, flavonol and phenolic acids profiles (non-anthocyanin phenolic compounds NAN) of 28 genotypes of Vitis vinifera L. grapes, from EVEGA germplasm bank, were defined in two consecutive vintages. To the best of our knowledge, for almost all the varieties under study, this is the first time that this characterization is reported.

  1. The description of the sample preparation and the analytical methods used require more detailed information. The grape samples were frozen at -20 degrees and then extracted with the extraction mixture. Were the fruits thawed, or was homogenization performed on frozen fruits? If the sample was thawed, how? 50 g of fruit were taken for extraction. How were the fruits divided to measure 50 g? Cut the fruit into quarters? So that the ratio of peel, pulp and seeds is the same as in the fruit? Was about 50 g of fruit weighed? This should be specified.

Response: Thank you for your detailed review and for helping us to make our research clearer for readers.

The weight of the samples (triplicate per variety) was to approximately 50 g, taking note of the exact weight of each sample. In order to clarify and make it clear in the manuscript, and specify a bit more as you recommend us, we explained the procedure as follows (Lines 148-153 of revised manuscript):

Polyphenolic substances of approximately 50 g of healthy, frozen whole grapes were extracted with 50 mL methanol/water/formic acid (50:48.5:1.5, v/v/v/v) according to the method described by Portu et al [37] with slight modifications. The methanol acid mixture was added to unfrozen gapes and homogenized (Moulinex 180 W grinder, Alençon, France), sonicated for 10 min at 50 Hz (Grant XUB5, Cambridge, England) and centrifuged at 5,000 rpm for 10 min (Allegra 25R, Beckman Coulter, Delaware, USA)

  1. I also recommend writing the gradient in the HPLC method, even if it is given in the cited publication.

Response: Thank you for your recommendation. We included the flow rate and linear solvents gradient in page 5, section 2.3.3. Identification and quantification (Lines 177-184 of the revised manuscript):

The following eluents and solvents gradient were used: (A) acetonitrile/water/formic acid, (3:88.5:8.5, v/v/v), (B) acetoni-trile/water/formic acid (50:41.5:8.5, v/v/v), and (C) methanol/water/formic acid (90:1.5:8.5, v/v/v), maintaining the column at 40 °C and the flow rate in 0.63 mL·min-1. The linear solvents gradient was as follows: 0 min, 96 % A, and 4 % B; 7 min, 96 % A, and 4 % B; 38 min, 70 % A, 17 % B, and 13 % C; 52 min, 50 % A, 30 % B, and 20 % C; 52.5 min, 30 % A, 40 % B, and 30 % C; 57 min, 50 % B, and 50 % C; 58 min, 50 % B, and 50 % C; 65 min, 96 % A, and 4 % B.

  1. The paper states that the identification of polyphenolic compounds was made on the basis of their elution order and by comparison with the retention times of commercially available standards. Of course, a more adequate tool to confirm the correct identification would be the use of LC-MS/MS, but why wasn't at least a comparison of the spectra with the available standards?

Response: Certainly, the identification by a LC-MS/MS is a good option. However, since the commercially standards were available, we carried out the identification: -by their elution order and -by comparison with the retention times of these commercially available standards. Moreover, we attached in a complementary file the calibration lines for each compound just in case you consider necessary to included them as supplementary material.

  1. I recommend to improve the readability of figure number 4. The legend is especially hard to read. A similar remark applies to charts 6-8. They are very unreadable.

Response: Thank you for your review and sorry for the inconvenient reading the figures. Quality has been improved in all figures, modifying their resolution or changing colours to make the data look clearer, we hope they are now much easier to read.

  1. Because one of the reasons for the research was to assess whether the wine obtained from a given variety will undergo co-pigmentation during storage, which will improve the color stability. Therefore, I believe that the conclusions should indicate a further direction of research related to the fermentation process of fruits with the highest content of polyphenolic compounds and check whether the actual color of such wines will be more stable.

Response: Thank you very much for your suggestions. A new sentence was included in the new version of the manuscript, on conclusion section Lines 629-633):

On the other hand, some NANV as ‘Corbillón’ ‘Evega 3’ and ‘Evega 6’ could be considered by their contents in flavanols and gallic acid in plurivarietal winemaking being a possible line of future research the analysis of these varieties´ wines in terms of evaluating if its color stability complies with the results obtained in this investigation.”

Round 2

Reviewer 1 Report

The manuscript entitled “Non anthocyanin compounds in minority red grapevine varieties traditionally cultivated in Galicia (Northwest Iberian Peninsula). Analysis of flavanols, flavonols and phenolic acids” should be accepted since the authors included all reviewer’s suggestions, which contributes to improve the quality of the manuscript.